# Optimal Hypothesis Selection in (Almost) Linear Time

**Maryam Aliakbarpour**
Department of Computer Science
Rice University
Houston, TX 77005
maryama@rice.edu

**Mark Bun**
Department of Computer Science
Boston University
Boston, MA 02215
mbun@bu.edu

**Adam Smith**
Department of Computer Science
Boston University
Boston, MA 02215
ads22@bu.edu

## Abstract

Hypothesis selection, also known as density estimation, is a fundamental problem in statistics and learning theory. Given a sample set from an unknown distribution $P$ and a finite class of candidate distributions (hypotheses) $\mathcal{H} := H_1, H_2, \ldots, H_n$, the goal is to design an algorithm that selects a distribution $\hat{H}$ from $\mathcal{H}$ that best describes $P$. The accuracy of the algorithm is measured by the distance between $\hat{H}$ and $P$, compared to the distance between the closest distribution in $\mathcal{H}$ and $P$ (denoted by OPT). Specifically, we aim for $\|\hat{H} - P\|_{\mathrm{TV}}$ to be at most $\alpha \cdot \mathrm{OPT} + \epsilon$ for some small $\epsilon$ and $\alpha$.

While the value of $\epsilon$ can be reduced with an increasing number of samples, $\alpha$ is an inherent characteristic of the algorithm. Achieving $\alpha < 3$ is impossible, even with only two candidate hypotheses, unless the number of samples is proportional to the domain size of $P$ [Bousquet, Kane, Moran '19]. Finding a computationally efficient algorithm that achieves the optimal $\alpha$ has been a primary focus of research since the early work of [Devroye, Lugosi '01]. Before our work, the algorithms achieving $\alpha < 5$ required time $\Omega(n^2)$. We present the first algorithm that operates in almost linear time ($\tilde{O}(n/\epsilon^3)$) and achieves $\alpha = 3$. This result improves upon a long line of hypothesis selection research. Previously known algorithms had either worse time complexity, a larger $\alpha$ factor, or additional assumptions about the problem setting. Additionally, we provide another (almost) linear-time algorithm with better dependency on the additive accuracy parameter $\epsilon$, albeit with a slightly worse accuracy parameter of $\alpha = 4$.

## 1  Introduction

Hypothesis selection, also known as density estimation, is a fundamental problem in statistics and learning theory. This problem involves identifying a density function that accurately represents the distribution of a given dataset. Suppose we are given a dataset of samples drawn from an unknown distribution $P$ and a finite class of known distributions, representing different hypotheses: $\mathcal{H} := \{H_1, H_2, \ldots, H_n\}$. The goal is to select a distribution in $\mathcal{H}$ that is close to $P$ in total variation (TV) distance.

Typically, learning a distribution over a domain $\mathcal{X}$ with an $\epsilon$ error in TV distance requires $\Omega(|\mathcal{X}|/\epsilon^2)$ samples, which presents a substantial lower bound in sample complexity for distributions over large

38th Conference on Neural Information Processing Systems (NeurIPS 2024).

domains. Surprisingly, the findings of Yatracos, Devroye, and Lugosi revealed that for hypothesis selection, this sample complexity can be independent of the domain size and only logarithmic in the number of hypotheses [Yat85, DL96, DL97]. With just $s := \Theta(\log n/\epsilon^2)$ samples from $P$, it is possible to learn $P$ within error $\alpha \cdot \mathsf{OPT} + \epsilon$, where $\mathsf{OPT}$ denotes the distance of the nearest distribution in $\mathcal{H}$ to $P$. Specifically, Devroye and Lugosi introduced two algorithms for this problem: the Scheffé tournament, which operates in $O(n^2 \cdot s)$ time[1] and achieves $\alpha = 9$; and the minimum distance estimate, which runs in $O(n^3 \cdot s)$ time and achieves $\alpha = 3$ [DL01, Chapter 6]. In these results, although $\epsilon$ can be decreased as the number of samples increases, $\alpha$ remains an inherent parameter of the algorithm.

Significant effort has been directed towards finding computationally efficient algorithms for this problem while maintaining sample efficiency. The trade-off between the accuracy parameter $\alpha$ and computational efficiency has been a focal point. Mahalanabis and Stefankovic in [MS08] enhanced the minimum distance estimate, improving the time complexity to $O(n^2 \cdot s)$. They also introduced a nearly linear-time algorithm that achieves $\alpha = 3$, but requires exponential time in $n$ for preprocessing the class $\mathcal{H}$. Other nearly linear-time algorithms were developed achieving $\alpha = 9$ in [AJOS14, AFJ+18, AAC+23][2] and $\alpha = 5$ in [ABS23]. Furthermore, linear-time algorithms have been presented under the assumption that the algorithm receives the value of $\mathsf{OPT}$ (or its upper bound) as input [DK14, ABS23]. However, achieving $\alpha < 3$ is not possible unless the number of samples becomes $\mathrm{poly}(|\mathcal{X}|)$, as indicated in [BKM19].

Despite the long-standing history of this problem, the following question remained open:

*Is there an algorithm for hypothesis selection with the optimal number of samples $s = O(\log(n)/\epsilon^2)$ and optimal accuracy parameter $\alpha = 3$ that runs in $O(n \cdot s)$ time?*

We present the first almost linear-time algorithm that uses the optimal number of samples and achieves the optimal accuracy parameter $\alpha = 3$. Our algorithm runs in $\tilde{O}(n \cdot s \ / \ \epsilon)$ time (see Theorem 2). Additionally, we introduce another algorithm with improved dependency on $\epsilon$, running in $\tilde{O}(n \cdot s)$ time while obtaining a slightly higher accuracy parameter, $\alpha = 4$ (see Theorem 3). Our results represent a significant step forward, as they are the first in decades to achieve time complexity linear in $n$ for any $\alpha < 5$. See Table 1.

**Applications of Hypothesis Selection**    The primary application of hypothesis selection is to identify the best distribution from a set of known models that represent potential underlying data distributions we can effectively manage. For example, this set might include Poisson, gamma, and binomial distributions with various parameters used to model the number of arrivals per time unit. This makes hypothesis selection applicable to tasks like interpretable distribution selection and strategy optimization, where the objective is to choose the most suitable model from available options.

Another key strength of hypothesis selection is its agnostic nature, allowing it to adapt even when the true distribution lies outside the considered class. This robustness makes it effective in noisy data settings, with applications in denoising and anomaly detection. From a theoretical standpoint, hypothesis selection is fundamental in learning structured distributions, particularly when combined with the cover method, as seen in learning mixtures of Gaussians [DK14, SOAJ14]. For additional references, see Section 1.3.

**Importance of improving $\alpha$ by a constant factor**    In most learning algorithms, the error guarantee decreases polynomially as the number of samples increases, so constant factors may not be as crucial. However, this is not the case in hypothesis selection. The output hypothesis is guaranteed to be $(\alpha \cdot \mathsf{OPT} + \epsilon)$-close to $P$ in total variation distance. While increasing the number of samples can reduce $\epsilon$ to negligible levels, it does not improve the term $\alpha \cdot \mathsf{OPT}$. Hence, $\alpha$ is an inherent property of the algorithm and directly impacts the best achievable error guarantee. Therefore, even a constant improvement in $\alpha$ is significant.

One might argue that, alternatively, $\mathsf{OPT}$ could be reduced by carefully curating the class $\mathcal{H}$. However, beyond the practical challenges of finding a better $\mathcal{H}$, it is unclear whether this can be achieved

---

[1]This time bound assumes constant-time comparisons of the $H_i$'s density functions. See Section 1.1.

[2]Theorem 4.1 in [AAC+23] states the result for $\alpha = 27$. However, according to personal correspondence with the authors, it is possible to modify their algorithm in conjunction with the minimum distance estimate and improve $\alpha$ to 9.

without significantly increasing the size of the hypothesis class $\mathcal{H}$. For example, in the cover method, when aiming to learn a distribution within a class $\mathcal{C}$, we set $\mathcal{H}$ to be a $\gamma$-net that serves as a cover for $\mathcal{C}$, ensuring that $\mathsf{OPT} < \gamma$. While using a finer $\gamma$-net can reduce $\mathsf{OPT}$, it may also drastically increase the size of $\mathcal{H}$, thereby increasing the algorithm's running time, since the size of the net can grow super-polynomially with respect to $\gamma$. For instance, in the case of mixtures of $k$ Gaussians, the size of the net depends on $\gamma$ as roughly $O(\gamma^{-3 \cdot k})$ (see [SOAJ14]). Thus, reducing $\gamma$ by a factor of three could exponentially increase the size of $\mathcal{H}$ in terms of $k$, thereby increasing both the running time and sample complexity, ultimately resulting in an inefficient algorithm.

Table 1: Summary of Past Results in Hypothesis Selection. All algorithms use $s = \Theta(\log n / \epsilon^2)$ samples.

| Result | $\alpha$ | Time Complexity | Additional requirement |
|---|---|---|---|
| Min distance estimate [DL01] | 3 | $O(n^3 \cdot s)$ | |
| Scheffé tournament [DL01] | 9 | $O(n^2 \cdot s)$ | |
| Min distance estimate [MS08] | 3 | $O(n^2 \cdot s)$ | |
| [AJOS14, AFJ$^+$18] | 9 | $\tilde{O}(n \cdot s)$ | |
| [ABS23] | 5 | $\tilde{O}(n \cdot s)$ | |
| [AAC$^+$23] | 9 | $O(n \cdot s / \log n)$ | |
| [MS08] | 3 | $O(n \cdot s)$ | Exponential time preprocessing |
| [DK14, ABS23] | $\geq 3$ | $\tilde{O}(n \cdot s)$ | Assume knowledge of $\mathsf{OPT}$ |
| Lower bound [BKM19] | | Achieving $\alpha < 3$ requires $\mathrm{poly}(|\mathcal{X}|)$ samples | |
| This work:Algorithm 1 | 3 | $\tilde{O}(n \cdot s \,/\, \epsilon)$ | |
| This work: Algorithm 4 | 4 | $\tilde{O}\left(n \cdot s\right)$ | |

## 1.1 Problem Setup

Suppose we have an *unknown* distribution $P$ over a domain $\mathcal{X}$ and a set of $n$ *known* distributions $\mathcal{H} := \{H_1, H_2, \ldots, H_n\}$. Let $\mathsf{OPT}$ denote the distance between $P$ and the closest distribution to it in $\mathcal{H}$ [3]: $\mathsf{OPT}(\mathcal{H}, P) := \min_{H \in \mathcal{H}} \|H - P\|_{\mathrm{TV}}$. We use the standard access model for this problem as introduced in [DL01]. The algorithm accesses the distributions through the following types of queries:

1. The algorithm can request a sample from the unknown distribution $P$.

2. The algorithm can compare the PDF of two known distributions: For every domain element $x \in \mathcal{X}$ and every pair of indices $i$ and $j$, it can ask if $H_j(x) < H_i(x)$. This is equivalent to asking if $x$ is in the Scheffé set of $H_i$ and $H_j$ (defined in Equation (2)).

3. The algorithm can query the probability mass of the Scheffé sets according to all the known distributions.

**Remark 1.** *The last requirement of our model can be relaxed. Only estimates of the probability masses of the Scheffé sets are needed for our algorithms. Thus, one can alternatively assume sample access to $H_i$'s, and estimate these values via samples.*

**Definition 1.1** (Proper learner for hypothesis selection). *Suppose algorithm $\mathcal{A}$ is given parameters $\epsilon, \delta \in (0,1)$, $\alpha \in \mathbb{R}_{>0}$ and has access to an unknown distribution $P$ and a class of $n$ known distributions $\mathcal{H}$ (as described above). We say $\mathcal{A}$ is an $(\alpha, \epsilon, \delta)$-proper learner for the hypothesis selection problem if for every $P$ and $\mathcal{H}$, $\mathcal{A}$ outputs $\hat{H} \in \mathcal{H}$ for which, with probability at least $1 - \delta$,*

$$\|\hat{H} - P\|_{TV} \leq \alpha \cdot \mathsf{OPT}(\mathcal{H}, P) + \epsilon. \tag{1}$$

---

[3]We omit the arguments $(\mathcal{H}, P)$ when they are clear from the context.

*We refer to $\alpha$ as the accuracy parameter, $\epsilon$ as the error (or proximity) parameter, and $\delta$ as the confidence parameter of the algorithm.*

## 1.2 Main theorems

Below, we provide informal versions of our theorems.

**Theorem 2.** *For every $\epsilon, \delta \in (0,1)$, Algorithm 1 is an $(\alpha = 3, \epsilon, \delta)$-proper learner for hypothesis selection that uses $s = O(\log(n/\delta)/\epsilon^2)$ samples and time $\tilde{O}(n \cdot s \: / \: (\delta^3 \epsilon)) = \tilde{O}(n/(\delta^3 \epsilon^3))$.*

**Theorem 3.** *For every $\epsilon, \delta \in (0,1)$, Algorithm 4 is an $(\alpha = 4, \epsilon, \delta)$-proper learner for hypothesis selection that uses $s = O(\log(n/\delta)/\epsilon^2)$ samples and time $\tilde{O}(n \cdot s \cdot \log(1/\delta)) = \tilde{O}(n \cdot \log^2(1/\delta)/\epsilon^2))$.*

For formal statements, see Theorem 5 (Appendix A) and Corollary B.1 (Appendix B).

Our results are the first to achieve time complexity linear in $n$ for any $\alpha < 5$. To achieve this, we introduce novel algorithmic techniques that will hopefully be broadly useful—see Section 3 for an overview. Both algorithms use the optimal number of samples. The first algorithm obtains optimal accuracy parameter $\alpha = 3$ with a time complexity of $\tilde{O}(n/(\delta^3 \epsilon) \cdot s)$, which is off by an $O(1/(\delta^3 \epsilon))$ factor. Our second algorithm achieves the optimal time complexity up to logarithmic factors while achieving a slightly higher $\alpha = 4$. Our results leave a fascinating open question: Can one combine the best aspects of both algorithms, maintaining $\alpha = 3$, achieving $O(n \cdot s)$ time (or even lower), and sample complexity $s = O(\log(n/\delta)/\epsilon^2)$?

**Remark 4.** *Readers may be surprised by the polynomial dependence on $1/\delta$ in Theorem 2. In many settings, the success probability of a learning algorithm can be amplified from a constant, say 2/3, to at least $1 - \delta$ at a cost of at most $\log(1/\delta)$ in running time and sample complexity. However, in hypothesis selection, there is no (known) general technique for boosting the confidence parameter while keeping $\alpha$ the same. The issue is that choosing the best output from several runs of a given algorithm requires executing a second hypothesis selection algorithm, which introduces another factor of $\alpha$ in the approximation—leading to a total factor of at least 9. As a result, these kinds of two-phase algorithms are not sufficient in the low $\alpha$ regime. Some previous results, such as [ABS23], also suffer from a polynomial dependency on $\delta$.*

## 1.3 Other related work

Hypothesis selection has been studied in various settings including improper setting. In [BKM19], the authors consider the improper version of the problem where the output hypothesis $\hat{H}$ may not necessarily be in $\mathcal{H}$. They presented an improper learner with accuracy guarantee $\alpha = 2$. The sample complexity of their algorithm was improved by [BBK+21], who gave an algorithm with nearly optimal sample complexity and the same accuracy parameter $\alpha = 2$. It is worth noting that our algorithms are proper learners and solve this problem with a slightly better sample complexity. In addition, like other proper learners, our algorithms select their output from a predefined set, which can facilitate choosing a distribution with specific structural property (e.g., mixture of Gaussians). In certain applications, this selection ensures consistency with the problem's underlying assumptions, which enhances interpretability and robustness.

The problem of hypothesis selection in *sublinear time* was studied for distributions on *discrete domains* [AAC+23]. Among other results, the authors developed a data structure that upon receiving samples from a unknown distribution $P$ returns a hypothesis $\hat{H}$ in $o(n)$ time. While their algorithm runs in sublinear time, their sample complexity depends on the domain size of the distribution, and their setting allows pre-processing of the class $\mathcal{H}$ in polynomial time. Another interesting variation for hypothesis selection is designing differentially private learners for the problem which has received attention over the past few years [BKSW19, CKM+19, GKK+20].

An important application of hypothesis selection arises when there is a structural assumption on the underlying distributions. One approach for learning these classes is to selectively choose a *cover* for the class. One can then use the learners for the standard hypothesis selection problem (which we study in the paper) and use the cover as the class $\mathcal{H}$. Examples of such structural assumptions include Poisson binomial distributions [DDS12], mixtures of Gaussians [KMV12, DK14, SOAJ14, DKS17, KSS18, ABM18, ABH+20], distributions with piecewise polynomial PDFs [ADLS17], and histograms [Pea95, CDSS14, CDKL22]. See Diakonikolas [Dia16] for a survey of results.

## 2 Preliminaries

**Notation:** We use the following notation throughout this article. We use $[n]$ to indicate the set $\{1, 2, \ldots, n\}$. For a distribution $Q$ over $\mathcal{X}$, $Q(x)$ denotes the PDF of $Q$ at the domain element $x \in \mathcal{X}$. For any measurable subset of the domain $S \subseteq \mathcal{X}$, $Q(S)$ indicates the probability mass of the set $S$ according to $Q$. We use $\|Q_1 - Q_2\|_{\text{TV}} := \sup_{S \subseteq \mathcal{X}} |Q_1(S) - Q_2(S)|$ to denote the total variation distance between $Q_1$ and $Q_2$. We say $Q_1$ is $\epsilon$-close to $Q_2$ if $\|Q_1 - Q_2\|_{\text{TV}}$ is at most $\epsilon$. Conversely, we say $Q_1$ is $\epsilon$-far from $Q_2$ if $\|Q_1 - Q_2\|_{\text{TV}}$ is greater than $\epsilon$. We use the standard $O, \Omega, \Theta$ notation for asymptotic functions. Additionally, we use $\tilde{O}, \tilde{\Omega}$, and $\tilde{\Theta}$ to hide polylog factors.

**Scheffé sets:** For every pair of hypotheses $H_i$ and $H_j$ in $\mathcal{H}$, we define the Scheffé set of $H_i$ and $H_j$ as follows:

$$\mathcal{S}_{i,j} := \left\{ \begin{array}{ll} \{x \in \mathcal{X} \mid H_i(x) < H_j(x)\} & \text{if } i \leq j \\ \mathcal{S}_{j,i} & \text{if } i > j \end{array} \right. \tag{2}$$

It is known that the Scheffé set maximizes the probability discrepancy between $H_i$ and $H_j$, thus fully characterizing the total variation distance between the two distributions:

$$\|H_i - H_j\|_{\text{TV}} = \sup_{S \subseteq \mathcal{X}} |H_i(S) - H_j(S)| = |H_i(\mathcal{S}_{i,j}) - H_j(\mathcal{S}_{i,j})| . \tag{3}$$

**The optimal hypothesis:** Recall that we assume the algorithm is given samples drawn from an unknown distribution $P$. Let $H_{i^*}$ denote the closest hypothesis in $\mathcal{H}$ to $P$. That is, $H_{i^*}$ is the hypothesis for which $\|H_{i^*} - P\|_{\text{TV}} = \text{OPT}$. When there is more than one hypothesis with this property, we pick one arbitrarily as $H_{i^*}$.

**Semi-distances:** For every pair $i, j$ in $[n]$, we define $w_j(H_i)$ to be $|H_i(\mathcal{S}_{i,j}) - P(\mathcal{S}_{i,j})|$. In words, $w_j(H_i)$ is the distance of $H_i$ to $P$ observed on the Scheffé set of $H_i$ and $H_j$. For every pair $i, j \in [n]$, we use $\hat{w}_j(H_i)$ to denote an estimate of $w_j(H_i)$ based on the observed sample. For the sake of consistency, we define $w_i(H_i)$ to be zero. In addition, we define the score function $W(H_i) := \max_{j \in [n]} w_j(H_i)$. Similarly, $\hat{W}(H_i)$ is defined to be $\max_{j \in [n]} \hat{w}_j(H_i)$.

**Refined access model:** Similar to previous work [DL01, MS08], we use estimates of the semi-distances. One can easily estimate these quantities, denoted by $\hat{w}_j(H_i)$, via the access model we have described earlier by letting $\hat{w}_j(H_i)$ be the empirical ratio of the samples that are in $\mathcal{S}_{i,j}$. Throughout this paper, we assume that there are two universal parameters $\delta' = \Theta(\delta)$ and $\epsilon' = \Theta(\epsilon)$, for which with probability $1 - \delta'$ every $\hat{w}_j(H_i)$ is within $\epsilon'$ of $w_j(H_i)$:

$$\forall i, j \in [n] : \quad |\hat{w}_j(H_i) - w_j(H_i)| \leq \epsilon'.$$

A simple application of the Hoeffding bound and the union bound shows that one can compute all of the estimates via $s = O(\log(n/\delta')/\epsilon'^2)$ samples, and each estimate can be computed in $O(s)$ time.

Our algorithms access the distributions in $\mathcal{H} \cup \{P\}$ only via querying $\hat{w}_j(H_i)$. This fact brings the sample complexity of our algorithms to $s = O(\log(n/\delta')/\epsilon'^2)$ samples. The time complexity of our algorithms is determined by the number of queries they make to the $\hat{w}_j(H_i)$'s, multiplied by time that we spend on each query, $O(s)$. Moreover, in the proofs of our theorems, we assume without loss of generality that all the $\hat{w}_j(H_i)$'s are accurate. Conditioning on the accuracy will not decrease the probability of correctness of any of our algorithms by more than $\delta'$ due to Fact C.1.

## 3 Overview of our techniques

In this section, we provide a high-level discussion of our algorithm. The important notations and definitions used here are provided in Section 2. For the high-level discussions in this section, we assume we have access to the exact values of the semi-distances $w_j(H_i)$. In the formal proofs presented in subsequent sections, we will substitute each $w_j(H_i)$ with an estimated value $\hat{w}_j(H_i)$.

If the error of these estimates is below $\epsilon' = \Theta(\epsilon)$, it can be shown that the overall impact of this substitution on our final distance guarantee (Equation 1) is at most $\Theta(\epsilon)$. See the "Refined access model" in Section 2 for further details.

### 3.1 Background: Semi-distances and the minimum distance estimate

To solve the hypothesis selection problem, we seek a *certificate* that ensures we output a hypothesis $\hat{H}$ such that $\|\hat{H} - P\|_{\mathrm{TV}}$ is at most $3\,\mathsf{OPT}$. Similar to previous work, our algorithms operate based on the probability masses of the *Scheffé sets* (Equation (2)) of pairs of hypotheses in $\mathcal{H}$. The semi-distance $w_j(H_i)$, defined as $|H_i(\mathcal{S}_{i,j}) - P(\mathcal{S}_{i,j})|$, captures the "distance" between $H_i$ and $P$ as measured on this particular set $\mathcal{S}_{i,j}$. One suggestion for readers to internalize the semi-distances is to view them as a distance between $H_i$ and $P$ that is measured from the perspective of $H_j$. By definition, $w_j(H_i)$ is always a lower bound for $\|H_i - P\|_{\mathrm{TV}}$:

$$w_j(H_i) := |H_i(\mathcal{S}_{i,j}) - P(\mathcal{S}_{i,j})| \leq \sup_{S \subseteq \mathcal{X}} |H_i(S) - P(S)| = \|H_i - P\|_{\mathrm{TV}}.$$

However, it is possible for $w_j(H_i)$ to be much lower, making it difficult for an algorithm to estimate the TV distance between $H_i$ and $P$ just using semi-distances.

Nevertheless, for each hypothesis $H_i$, a specific semi-distance $w_{i^*}(H_i)$, associated with the optimal hypothesis $H_{i^*}$, determines its quality. As shown in the following proof, if $w_{i^*}(H_i) \leq \mathsf{OPT}$, then $H_i$ is $3\,\mathsf{OPT}$-close to $P$, with the total variation distance between $H_i$ and $P$ bounded by $w_{i^*}(H_i)$ and $\mathsf{OPT}$ via the triangle inequality:

$$\|H_i - P\|_{\mathrm{TV}} \leq \|H_i - H_{i^*}\|_{\mathrm{TV}} + \|H_{i^*} - P\|_{\mathrm{TV}} = |H_i(\mathcal{S}_{i,j}) - H_j(\mathcal{S}_{i,j})| + \mathsf{OPT} \quad \text{(By Eq. 3)}$$
$$\leq w_{i^*}(H_i) + w_i(H_{i^*}) + \mathsf{OPT} \leq w_{i^*}(H_i) + 2\,\mathsf{OPT}.$$

This observation implies that if we assert that $w_{i^*}(H_i)$ is bounded by $\mathsf{OPT}$, we can output $H_i$ as our final solution to the problem and be done. The challenge is that we neither know $i^*$ nor $\mathsf{OPT}$.

This issue is addressed by the *minimum distance estimate* presented in [DL01, MS08] using a score function $W(H_i)$, as defined earlier: $W(H_i) := \max_{j \in [n]} w_j(H_i)$. The minimum distance estimate outputs a hypothesis $\hat{H}$ that minimizes $W(H_i)$. We can assert that, for the output of this algorithm, $\hat{H}$, $w_{i^*}(\hat{H})$ is at most $\mathsf{OPT}$. This approach simultaneously bypasses the issues of not knowing $i^*$ and $\mathsf{OPT}$.

Note that $W(H_i)$ serves as a proxy for the quality of $H_i$ since $W(H_i)$ is an upper bound for $w_{i^*}(H_i)$. Using $W(H_i)$, we can address the issue of not knowing $i^*$. On the other hand, although $\mathsf{OPT}$ is not known, we have a good lower bound for $\mathsf{OPT}$, which is $W(H_{i^*})$. Putting these observations together, we obtain:

$$\|\hat{H} - P\|_{\mathrm{TV}} \leq w_{i^*}(\hat{H}) + 2\,\mathsf{OPT} \leq W(\hat{H}) + 2\,\mathsf{OPT} \leq W(H_{i^*}) + 2\,\mathsf{OPT} \leq 3\,\mathsf{OPT}.$$

These inequalities are derived using the triangle inequality and the fact that $\hat{H}$ was chosen to be the $\arg\min_{i \in [n]} W(H_i)$. See Figure 1 for an illustration of the above inequality.

The primary hurdle with the minimum distance estimate is that it is costly to compute. Computing each $W(H_i)$ takes $O(n \cdot s)$ time, bringing the total time complexity of the algorithm to $O(n^2 \cdot s)$. One might naturally conjecture that sampling may help to compute an estimate of $W(H_i)$. Instead of using $W(H_i) := \max_{j \in [n]} w_j(H_i)$, we can use $\tilde{W}(H_i) := \max_{j \in R} w_j(H_i)$ where $R$ is a set of random indices in $[n]$. The issue with this approach is that there is no guarantee of $i^*$ being selected in $R$, making $\tilde{W}(H_i)$ too low while $H_i$ may be far from $P$. Hence sampling, if used in a trivial manner, is not beneficial.

### 3.2 The algorithm with $\alpha = 3$

In this section, we present an overview of Algorithm 1 that attains $\alpha = 3$. The details of this algorithm and its related theorems are provided in Section A. At a high level, similar to the minimum distance estimate, we work towards finding a hypothesis that minimizes $W(H_i)$. To increase efficiency, we

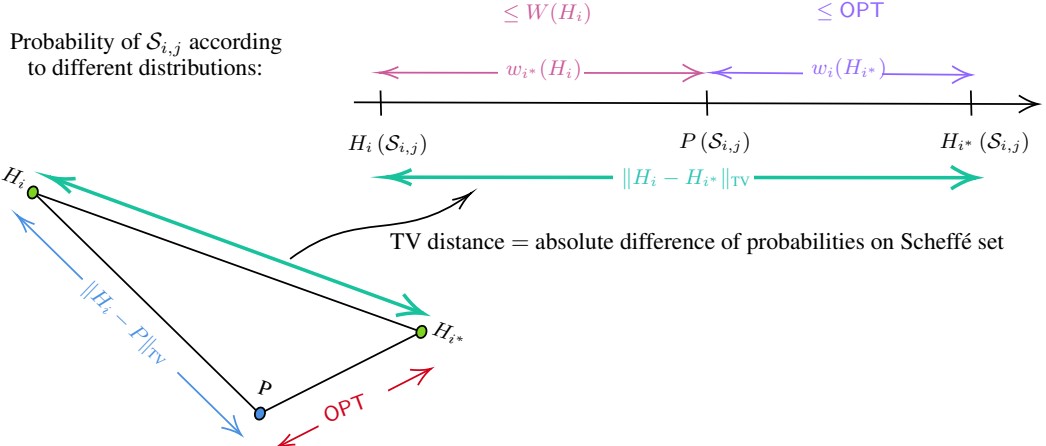

Figure 1: A description of bounding $\|H_i - P\|_{\mathrm{TV}}$ via $W(H_i)$ and OPT

work with estimates of $W(H_i)$'s, denoted by $\tilde{W}(H_i)$. The general structure of our algorithm is as follows: Initially, all $\tilde{W}(H_i)$ are set to zero. At every step, we come up with several pairs of hypotheses $H_i$ and $H_j$ and update our estimates by setting $\tilde{W}(H_j)$ to $\max(\tilde{W}(H_j), w_i(H_j))$. Our approach ensures that at every step, $\tilde{W}(H_i)$ is equal to $\max_{j \in R} w_j(H_i)$ for a small, carefully chosen set $R$. Eventually, we output a hypothesis with roughly the smallest $\tilde{W}(H_i)$.

**Focusing on small $\tilde{W}$:**  Our first idea is to focus on updating the $\tilde{W}$ for hypotheses whose $\tilde{W}(H_i)$ values are around the current minimum $\tilde{W}$. The rationale for this action comes from a simple fact: $\tilde{W}(H_j)$ always underestimates the value of $W(H_j)$. Hence, if we observe that $\tilde{W}(H_i)$ is substantially larger than the current minimum, then $W(H_i)$ is also substantially larger than the current minimum. This implies that $H_i$ is not a suitable candidate for the minimum at this stage of the algorithm, and it can be ignored for now.

**Bucketing hypotheses based on $\tilde{W}$:**  To implement this idea, we partition the hypotheses into $k = \Theta(1/\epsilon')$ buckets $\mathcal{B} := \{B_1, B_2, \ldots, B_k\}$ based on $\tilde{W}(H_i)$. The bucket $\ell$ contains all the hypotheses $H_i$ such that $\tilde{W}(H_i) \in [(\ell-1)\epsilon', \ell\epsilon')$. At every step, we focus on the smallest non-empty bucket $B_\ell$ (the smallest $\ell$ for which $|B_\ell| \neq 0$). $B_\ell$ contains all the hypotheses whose $\tilde{W}(H_j)$ is around the minimum $\tilde{W}$. We pick pairs of hypotheses, $H_i \in \mathcal{H}$ and $H_j \in B_\ell$, and update $\tilde{W}(H_j)$. Note that our updates may increase $\tilde{W}(H_j)$, and we may remove $H_j$ from $B_\ell$ and put it into a larger bucket (a bucket with a larger index $\ell$). Also, observe that we never move a hypothesis into a smaller bucket since $\tilde{W}(H_j)$ never decreases. We continue our updates to reach one of the following outcomes:

- $B_\ell$ becomes empty. That is, our updates made all $\tilde{W}(H_i)$ fall out of the range of these buckets $[0, \ell\,\epsilon')$. Thus, every $\tilde{W}(H_i)$ (and consequently every $W(H_i)$) is at least $\ell \cdot \epsilon'$. Every time that we empty out a bucket, we have increased our threshold for minimum $W(H_i)$ by $\epsilon'$. Hence, we are getting closer to a bucket with the true minimum, which we hope to reach in $O(1/\epsilon')$ steps.
- $B_\ell$ is not empty, but we can confidently confirm most of the hypotheses in $B_\ell$ are an acceptable output for the algorithm. Although we cannot ensure that $H_{i^*}$ is indeed in $B_\ell$, we can find an acceptable final answer by selecting a random hypothesis in $B_\ell$.

**Which pairs to update?**  Next, we outline our update scheme to implement the above ideas in linear time. To enhance time efficiency, our aim is to optimize the updating process to ensure both *quality* and *quantity* in the chosen updates. Quality, in this context, relates to the extent of change in $\tilde{W}(H_j)$ following an update: We consider $H_i$ to have made a *substantial update* to $H_j$ if $\tilde{W}(H_j) + \epsilon' < \hat{w}_j(H_i)$. Such updates cause a significant shift, increasing the value of $\tilde{W}(H_j)$ by

more than $\epsilon'$ and subsequently moving $H_j$ to a different bucket with a higher index $\ell$. We refer to this event as $H_i$ *removing* $H_j$ from its bucket. The quantity, on the other hand, relates to the number of $H_j$ instances that a given $H_i$ can remove from $B_\ell$. An ideal $H_i$ eliminates a substantial portion of hypotheses from $B_\ell$ (say a constant fraction). We call such an $H_i$ a *prompting hypothesis*. Now, if for $O(\log |B_\ell|)$ rounds we find a prompting hypothesis and update all the $\tilde{W}(H_j)$ for $H_j \in B_\ell$, we will empty out the bucket $B_\ell$, and we can move forward.

**Finding a prompting hypothesis:**    To find a prompting hypothesis quickly, we iterate over all $H_i$ in $\mathcal{H}$, sample a few $H_j$, and check if $H_i$ substantially updates $\tilde{W}(H_j)$. If $H_i$ substantially improves a large fraction of the sampled hypotheses, then we declare that $H_i$ is a prompting hypothesis. See Section A.1.1 for further details. In addition to that, we provide a more advanced version of this procedure in Section A.1.2 that allows us to shave off an $O(\log n)$ factor.

**Getting stuck? Here is your way out:**    What happens when $B_\ell$ is not empty, and we cannot find a prompting hypothesis? We show that if we do not find a prompting hypothesis, then a random hypothesis in $B_\ell$ is an acceptable answer.

The surprising part about this statement is that it holds regardless of the size of the bucket $B_\ell$ due to hypothesis sampling procedure we have to find a prompting hypothesis. In search for a prompting hypothesis, we iterate over all $H_i$ and sample roughly $O(\log n/\delta)$ many hypotheses $H_j$'s in $B_\ell$. We check, if $H_i$ can remove them from the bucket. Note that if $H_i$ was not found to be prompting, it implies that $H_{i*}$ cannot substantially update most of the hypotheses in $B_\ell$. Thus, We have that with high probability for $1 - \delta$ fraction of the hypotheses in $B_\ell$, $w_i(H_j) \le \epsilon' \cdot \ell$.

The clever hack here is an observation about $H_{i*}$. Earlier, we discussed that we are looking for a hypothesis $H_i$ with $w_{i*}(H_i) \le \mathsf{OPT}$. We claim that a random hypothesis in the last $B_\ell$ (almost) has this property. On one hand, given that $H_{i*}$ was not found to be a prompting hypothesis, for $1 - \delta$ fraction of $H_i$ in $B_\ell$, $w_{i*}(H_i)$ must be at most $\ell \cdot \epsilon'$. On the other hand, the fact that all the previous buckets, $B_1, \ldots, B_{\ell-1}$, are empty indicates $H_{i*}$ has shown a semi-distance of at least $(\ell - 1)\,\epsilon'$. Hence, $\mathsf{OPT}$, which is at least as large as all $H_{i*}$ semi-distances, is at least $(\ell - 1)\,\epsilon'$. Therefore, $w_{i*}(H_i) \le \epsilon' + \mathsf{OPT}$. Therefore, for $1 - \delta$ fraction of the hypotheses in $B_\ell$, they are $3\,\mathsf{OPT} + \Theta(\epsilon)$ close to $P$. For a formal argument, see Lemma A.1.

Now, assume the search for the prompting hypothesis fails. Recall that during the search, we have checked every single hypothesis in $\mathcal{H}$. During the search, at some point, we must have stumbled upon $H_{i*}$ and did not declare it as a prompting hypothesis. In this case, either our sampling for substantial updates failed (which happens with small probability), or we can infer that there are not too many far hypotheses in $B_\ell$. Either way, we can output a random hypothesis from $B_\ell$ as the final sample without increasing the error probability by too much.

**Dependency on $\delta$:**    It is worth noting that this last step results in a polynomial dependency on $\delta$ (as opposed to a more desirable dependency of $\log(1/\delta)$). This is mainly due to the fact that to ensure that a random hypothesis in $B_\ell$ is not too far, with a probability of $1 - \delta$ in a single round, we have to try $O(\log(n)/\delta)$ hypotheses in $B_\ell$ and check if $H_{i*}$ is prompting them. Hence, relying on this structural property of $H_{i*}$ makes a polynomial dependency on $\delta$ inevitable. Improving the dependency on $\delta$ for this algorithm would require new algorithmic ideas.

### 3.3    The algorithm with $\alpha = 4$

We introduce another (almost) linear-time algorithm for the hypothesis selection problem, where the time complexity shows improved dependency on $\epsilon$. However, this algorithm has a slightly worse accuracy parameter compared to our previous algorithm: $\alpha = 4$. This algorithm and the subsequent theorems are provided in Section B.

**Algorithm with a guessed upper bound of $\mathsf{OPT}$:**    As mentioned earlier, unlike previous work, our algorithm does not receive any prior information about the value of $\mathsf{OPT}$. It might be speculated that there exists an easy reduction between our problem and another version of hypothesis selection, where an auxiliary parameter $\sigma$ is provided to the algorithm such that $\mathsf{OPT} \le \sigma$. A learner without the knowledge of $\sigma$ can make a guess about $\sigma$ and run one of the existing algorithms that works with the knowledge of an upper bound of $\mathsf{OPT}$ (e.g., [ABS23]) and check if it finds a good hypothesis or

not. With this procedure in mind, we can run a binary search to find the smallest $\sigma$ for which we find a hypothesis.[4] However, a challenge with this approach is that the algorithm might yield a hypothesis even when $\sigma <$ OPT, making it difficult for us to *refute* the hypothesis that is found. Even if the output $\hat{H}$ satisfies $W(\hat{H}) > \sigma$, it does not necessarily invalidate our guessed value $\sigma$.

To overcome this challenge, we design an algorithm, namely $\mathcal{A}$, with enhanced accuracy guarantees for the output hypothesis. We provide $\mathcal{A}$ with the value $\sigma$. We treat $\sigma$ as a "guess" for which we hope that OPT $\leq \sigma$. We require the algorithm to either refute our guess and declare OPT $> \sigma$, or output a hypothesis $\hat{H}$ with a reasonable distance to $P$, irrespective of the relationship between OPT and $\sigma$. More precisely, the distance of the output hypothesis should be bounded by: $\|\hat{H} - P\|_{\mathrm{TV}} \leq \alpha \cdot \max(\mathsf{OPT}, \sigma) + \epsilon$. With these revised guarantees, it is permissible to run a binary search over different values of $\sigma$ in $\{\epsilon, 2\epsilon, \ldots, \lceil 1/\epsilon \rceil \, \epsilon\}$, and output the hypothesis that $\mathcal{A}$ returns on the smallest $\sigma$. For the rest of this discussion, we focus on designing $\mathcal{A}$ and a provided parameter $\sigma$.

**Seeds and finding an acceptable hypothesis via seeds:**   In this algorithm, we introduce a new concept called a *seed* for hypotheses. A seed provides us with a structural property that enables us to identify an acceptable hypothesis, regardless of the relationship between $H_{i^*}$ and the seed. More formally, for a hypothesis $H_i$, we define $\mathcal{M}_{a,b}(H_i)$ as the set of hypotheses $H_j$ such that $w_j(H_i)$ is between $a$ and $b$:

$$\mathcal{M}_{a,b}(H_i) := \{H_j \in \mathcal{H} \mid a < \hat{w}_j(H_i) \leq b\} \, .$$

We may omit the subscript of $\mathcal{M}$ when it is clear from the context. We say a hypothesis $H_i$ is a seed if almost all of its $\hat{w}_j(H_i)$ are low and its $W(H_i)$ is not too large. The quality of a seed is determined by three parameters: $a$, $b$, and $m$.

**Definition 3.1** (Seed). *Given parameters $a, b \in \mathbb{R}_{\geq 0}$, and a non-negative integer $m$, we say a hypothesis is an $(a, b, m)$-seed if the following hold: 1) $\hat{W}(H_i) \leq b$, and 2) $|\mathcal{M}_{a,b}(H_i)| \leq m$.*

A seed with a small-sized set $\mathcal{M}$ and a small $b$ exhibits an interesting property: Suppose we have a seed $H_i$ with a constant-sized set $\mathcal{M}$. In this case, in $O(n)$ time, we can compute $W(H_j)$ of every hypothesis $H_j$ in $\mathcal{M}$. Now, let $H_\ell \in \mathcal{M}$ be one of the hypotheses that minimize $W(H_j)$: $H_\ell := \arg\min_{H_j \in \mathcal{M}} W(H_j)$. We claim either $H_i$ or $H_\ell$ is acceptable output in this case.

When $H_{i^*}$ is in $\mathcal{H} \setminus \mathcal{M}$, we can infer that $w_{i^*}(H_i)$ is at most $b$ (or, more precisely, $b + \epsilon'$). From our earlier discussion, if $b$ is sufficiently small (compared to $\sigma$), we can show that $H_i$ is close to $P$. In cases where $H_{i^*}$ is in $\mathcal{M}$, although we do not have strong guarantees for $H_i$ itself, we have a very good answer already: $H_\ell$ is essentially a minimum distance estimate for the set $\mathcal{M}$ which includes $H_{i^*}$. Thus, we can conclude: First, $H_\ell$ is an acceptable output because it is 3 OPT-close; second, $W(H_\ell)$ is a lower bound for OPT.

Now, as we described, we have two acceptable outputs for two cases of the problem. For $H_{i^*} \in \mathcal{H} \setminus \mathcal{M}$, $H_i$ is an acceptable answer. For $H_{i^*} \in \mathcal{M}$, $H_\ell$ is an acceptable answer. The only problem is that we do not know which case we are in. Our strategy is to pick a hypothesis that is still reasonably close to $P$ even if we are in the other case. Let us fix a threshold value $T$. We compare $W(H_\ell)$ with $T\sigma$: 1) If $W(H_\ell) \leq T \cdot \sigma$, as we have discussed earlier, $H_\ell$ is $(T \cdot \sigma + 2\mathsf{OPT})$-close to $P$. This bound holds regardless of where $H_{i^*}$ is. 2) Next, if $W(H_\ell) > T \cdot \sigma$, we cannot say $H_\ell$ is a good choice for us. However, we can conclude that $H_i$ is a close hypothesis to $P$ even when $H_{i^*}$ happens to be in $\mathcal{M}$. In this case, we know that OPT $\geq W(H_\ell) > T \cdot \sigma$. We take advantage of this knowledge and obtain the following bound:

$$\|H_{i^*} - P\|_{\mathrm{TV}} = b\,\sigma + 2\,\mathsf{OPT} \leq \left(\frac{b}{T} + 2\right) \mathsf{OPT} \, .$$

Thus, regardless of where $H_{i^*}$ is, $H_i$ is $(\max(a, b/T) + 2) \cdot \max(\sigma, \mathsf{OPT})$. Now, the algorithm is fairly straightforward. If $W(H_\ell)$ is small, output $H_\ell$; otherwise, output $H_i$. We summarize these four cases in the following table. Depending on $a$ and $b$, we set $T$ to minimize the maximum distance we

---

[4]This idea works to some extent, but it causes $\alpha$ to be increased by 2, which is not useful in our setting. See [ABS23] for more details.

endure in these cases. It is worth noting that in this argument, we did not rely on any prior knowledge concerning the relationship between $\sigma$ and $\mathsf{OPT}$.

| | $H_{i^*} \in \mathcal{M}$ | $H_{i^*} \in \mathcal{H} \setminus \mathcal{M}$ |
|---|---|---|
| $W(H_\ell) > T \cdot \sigma$ | $\|H_i - P\|_{\mathrm{TV}} \le a \cdot \sigma + 2\,\mathsf{OPT}$ | $\|H_i - P\|_{\mathrm{TV}} \le (b/T + 2)\mathsf{OPT}$ |
| $W(H_\ell) > T \cdot \sigma$ | $\|H_\ell - P\|_{\mathrm{TV}} \le T \cdot \sigma + 2\,\mathsf{OPT}$ | $\|H_\ell - P\|_{\mathrm{TV}} \le T \cdot \sigma + 2\,\mathsf{OPT}$ |

Table 2: Four cases when we process a good seed.

**How to find the first seed?** Here, we provide an overview of our approach for identifying an initial seed. For a detailed explanation of the algorithm and its performance, refer to Section B.1.

To start, fix two parameters, $a = \sigma$ and $b = 3\sigma$. To identify a seed hypothesis, we iterate over all hypotheses $H_1, \ldots, H_n$, checking whether each hypothesis $H_i$ is a strong seed (i.e., a seed with a small $|\mathcal{M}|$) by sampling several $H_j$'s and verifying if $w_j(H_i)$ is at most $a$. Roughly speaking, for an integer $m \in [n]$, if we sample approximately $\tilde{O}(n/m)$ $H_j$'s and find that no $w_j(H_i) > a$, we can, with high probability, confirm that the size of $\mathcal{M}$ does not exceed $m$. This approach requires $O(n^2/m)$ time.

There are, however, a few caveats with this method. First, if a seed is identified using the above approach, it may have a large $W(H_i) > b$. In this case, we can infer that $H_i$ is likely far from $P$. Given the time invested in identifying $H_i$, we aim to leverage this information. Broadly, observing that $w_j(H_i) \le a$ implies that many hypotheses are close to $H_i$ (assuming that $w_i(H_j)$ values are also small). Knowing that $H_i$ is far from $P$, we can *mark* all hypotheses close to $H_i$ as also far, allowing us to proceed with the search. While this may seem counterproductive, marking a significant number of hypotheses as "far" constitutes progress for our algorithm. Specifically, if, at some point, all hypotheses are marked, we can declare that $\sigma < \mathsf{OPT}$.

The second caveat is more challenging. Ideally, we seek a seed with a constant $m$, but finding such a seed would require $\tilde{O}(n^2/m) = \tilde{O}(n^2)$ time. Consequently, for a linear-time algorithm, we can only afford to find seeds where $m = \Theta(n)$. In other words, the quality of the seed we can initially identify is much lower than the quality required for a producing solution. This leads to our next idea: boosting a seed, which is an approach to incrementally improve the seed's quality in roughly $O(\log n)$ steps.

**Boosting a Seed:** In this process, we use an initial seed to iteratively find a stronger seed with a reduced value of $|\mathcal{M}|$. For a formal argument, see Section B.2. Assume a rate parameter $\eta \in (0,1)$. As discussed earlier, by spending $O(n/\eta)$ time, we can identify a seed with $m \approx \eta \cdot n$. Initially, $m = |\mathcal{M}|$ might be $O(n)$. Hence, rather than calculating $W(H_j)$ exactly for all $H_j \in \mathcal{M}$, we compute an approximate maximum semi-distance $\tilde{W}(H_j)$ by sampling $t$ hypotheses. Specifically, for each $H_j \in \mathcal{M}$, we set $\tilde{W}(H_j) := \max_{H_k} \hat{w}_k(H_j)$, where $H_k$'s are sample hypotheses. Using these approximate values $\tilde{W}$, we process the seed as follows. Let $H_\ell$ denote the hypothesis minimizing $\tilde{W}(H_j)$, with the following possible outcomes:

1. High $\tilde{W}(H_\ell)$: If $\tilde{W}(H_\ell)$ is high, all $\tilde{W}(H_j)$ (and thus $W(H_j)$) values are likely large, making $H_i$ an acceptable final solution.

2. Low $\tilde{W}(H_\ell)$: While this does not guarantee a low $W(H_\ell)$, it suggests that most $w_i(H_\ell)$ values are small. Based on the value of $W(H_\ell)$, we consider two cases:

   2.1 $W(H_\ell) \gg \tilde{W}(H_\ell)$: $H_\ell$ is likely far from $P$ but has many close hypotheses, allowing us to mark these nearby hypotheses as far.
   2.2 Moderate $W(H_\ell)$: In this case, $H_\ell$ can serve as our new seed, as it yields a smaller set $\mathcal{M}(H_\ell)$ than $H_i$.

Note that when we select $H_\ell$ as our new seed, $\mathcal{M}(H_\ell)$ is roughly $O(n/t)$. However, we compute $\tilde{W}(H_j)$'s for only $|\mathcal{M}(H_i)| \approx \eta \cdot n$ many hypotheses. Hence, with an increased sample size $t = O(1/\eta^2)$, this approach yields a smaller $|\mathcal{M}(H_\ell)| \approx \eta^2 \cdot n$. The step of the process requires $O(|\mathcal{M}| \cdot t) = m/\eta^2 = O(n/\eta)$, paralleling the initial search. By repeating, we progressively reduce $m = |\mathcal{M}|$, and for a constant $\eta$, $m$ decreases by a fixed factor until reaching a constant $m$ seed, as desired.

## Acknowledgments

M.A. was supported by NSF awards CNS-2120667, CNS-2120603, CCF-1934846, and BU's Hariri Institute for Computing. This work was initiated while M.A. was affiliated with Boston University and Northeastern University and was done in part while M.A. was a research fellow at the Simons Institute for the Theory of Computing. M.B. was supported in part by NSF award CNS-2046425 and a Sloan Research Fellowship. A.S. was supported in part by NSF awards CCF-1763786 and CNS-2120667 as well as Faculty Awards from Google and Apple.

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

**Algorithm 1** An algorithm for hypothesis selection with $\alpha = 3$.

1: **procedure** SELECT-HYPOTHESIS($\mathcal{H}$, $\epsilon$, $\delta$, and query access to $\hat{w}_j(H_i)$'s)
2:      $\tilde{W}(H_i) \leftarrow 0$ for every $i \in [n]$
3:      $k \leftarrow \lfloor 1/\epsilon' \rfloor + 1$
4:      $B_1 \leftarrow \mathcal{H}$
5:      $B_2, \ldots, B_k \leftarrow \emptyset$
6:      $\epsilon' \leftarrow \epsilon/3$
7:      $\delta' \leftarrow \delta/3$
8:      $\gamma \leftarrow \delta'$
9:      $\delta_{fn} \leftarrow \delta'/k$
10:     $\delta_{fp} \leftarrow \gamma \cdot \delta'/(4\,k\,n\,\log(n))$
11:     **for** $\ell = 1, \ldots, k$ **do**                          ▷ Iterating over buckets
12:        **for** $i = 1, \ldots, n$ **do**          ▷ Iterating over hypotheses to find a prompting one
13:           **if** $|B_\ell| = 0$ **then**
14:              Break the "for" loop and move on to the next bucket.
15:           **if** IS-PROMPTING($\mathcal{B}$, $\ell$, $H_i$, $\gamma$, $\epsilon'$, $\delta'$) **then**      ▷ Checks if $H_i$ is prompting.
16:              $C_{\ell i} \leftarrow 0$     ▷ A counter for the number of hypotheses that $H_i$ removes from $B_\ell$
17:              **for** $H_j \in B_\ell$ **do**     ▷ Update all hypotheses in $B_\ell$ via the prompting hypothesis
18:                 **if** $\hat{w}_i(H_j) > \tilde{W}(H_j) + \epsilon'$ **then**
19:                    $\tilde{W}(H_j) \leftarrow \max\left(\tilde{W}(H_j), \hat{w}_i(H_j)\right)$
20:                    Move $H_j$ to $B_{\lceil \tilde{W}(H_j)/\epsilon' \rceil}$
21:                    $C_{\ell i} \leftarrow C_{\ell i} + 1$
22:              **if** $\frac{C_{\ell i}}{C_{\ell i} + |B_\ell|} < \gamma/2$ **then**        ▷ It indicates $H_i$ was not prompting.
23:                 Output $\perp$ and halt.
24:              $i \leftarrow 0$                               ▷ Restarts the "for" loop.
25:        **if** $|B_\ell| \neq 0$ **then**
26:           $\hat{H} \leftarrow$ a random hypothesis in $B_\ell$
27:           Output $\hat{H}$ and halt.

## A   Almost linear time algorithm with $\alpha = 3$

In this section, we focus on our first result, the algorithm with $\alpha = 3$. The pseudocode of our approach is presented in Algorithm 1. We prove the performance of this algorithm in Theorem 5. An overview of our approach is described in Section 3.2.

At a high level, our algorithm begins with all $\tilde{W}(H_i)$ set to zero, and all hypotheses placed in $B_1$. We then iterate over all buckets. In round $\ell$, we check if there is a prompting hypothesis $H_i \in \mathcal{H}$ that can move a significant fraction of hypotheses from $B_\ell$. Specifically, we look for an $H_i$ that moves a $\gamma$-fraction of hypotheses from $B_\ell$ by updating them, where $\gamma = \Theta(\delta)$. If we find such a hypothesis $H_i$, we update all the hypotheses in $B_\ell$ and move as many as possible out of $B_\ell$. During this process, we count the number of hypotheses that $H_i$ removes from $B_\ell$, which is tracked by $C_{\ell i}$ in the algorithm. If the actual fraction of hypotheses removed from $B_\ell$ is less than $\gamma/2$, we infer that IS-PROMPTING falsely declared $H_i$ as a prompting hypothesis (false positive). Thus, the algorithm returns $\perp$. Otherwise, we restart the search by setting $i = 1$ and iterate over all $H_i$ to find the next prompting hypothesis. We continue this process until no prompting hypothesis is found or the bucket is emptied. If we completely empty a bucket, we move on to the next one. If not, we select a random hypothesis in $B_\ell$ as the final output and halt.

**Theorem 5.** *Suppose $\mathcal{H}$ is a class of $n$ known hypotheses and $P$ is an unknown hypothesis. Assume the algorithm has access to accurate estimates of $\hat{w}_j(H_i)$'s that have an error of at most $\epsilon'$. Let $\epsilon$ and $\delta$ be two parameters in $(0, 1)$. If $3\,\epsilon' \leq \epsilon$, then Algorithm 1 is an $(\alpha = 3, \epsilon, \delta)$-proper learner for $P$ in $\mathcal{H}$. The running time of our algorithm is $\tilde{O}((n/(\delta^3 \epsilon)) \cdot T_q)$, where $T_q$ is the time to obtain each $\hat{w}_j(H_i)$.*

*Proof.* Our proof consists of three parts:

1. If the algorithm returns $\hat{H}$, with high probability, $\hat{H}$ is not too far from $P$.

2. While the algorithm may not produce a hypothesis all the time, the overall failure probability of the algorithm is low.

3. The running time of the algorithm meets the desired bounds in the theorem.

**The output $\hat{H}$ is close with high probability.** Suppose we output $\hat{H} \in B_\ell$ at step $\ell$. Our goal is to show that $\hat{H}$ is $(3\,\mathsf{OPT} + \epsilon)$-close to $P$ with probability $1 - \delta'$. We define two events $\mathcal{E}_\ell$ and $\overline{\mathcal{E}_\ell}$ for each $\ell \in [k]$ based on the fraction of far hypotheses in $B_\ell$ in Line 25:[5]

$\mathcal{E}_\ell :=$ the event that a non-empty $B_\ell$ in Line 25 contains at least $\lceil \gamma \cdot |B_\ell| \rceil$
  hypotheses that are $(3\,\mathsf{OPT} + \epsilon)$-far from $P$.

$\overline{\mathcal{E}_\ell} :=$ the event that a non-empty $B_\ell$ in Line 25 contains fewer than $\lceil \gamma \cdot |B_\ell| \rceil$
  hypotheses that are $(3\,\mathsf{OPT} + \epsilon)$-far from $P$.

Conditioning on the event $\overline{\mathcal{E}_\ell}$, fewer than a $\gamma$ fraction of the hypotheses in $B_\ell$ are far. Since $\hat{H}$ is a randomly selected hypothesis in $B_\ell$, the probability of $\hat{H}$ being far is less than $\gamma$:

$$\mathbf{Pr}\Big[ \|\hat{H} - P\|_{\mathrm{TV}} > 3\,\mathsf{OPT} + \epsilon \mid \overline{\mathcal{E}_\ell} \Big] < \gamma. \tag{4}$$

Next, we focus on the case where $\mathcal{E}_\ell$ holds: at least a $\gamma$ fraction of the hypotheses in $B_\ell$ are $(3\,\mathsf{OPT} + \epsilon)$-far from $P$. In the following lemma, we show that $H_{i^*}$ is a prompting hypothesis for such a $B_\ell$. The proof of this lemma is available in Section A.2.

**Lemma A.1.** *Suppose $B_\ell \subseteq \mathcal{H}$ is the smallest non-empty bucket for which at least a $\gamma$ fraction of its hypotheses are $(3\,\mathsf{OPT} + \epsilon)$-far from $P$. Then $H_{i^*}$ is a prompting hypothesis that can remove a $\gamma$ fraction of the hypotheses in $B_\ell$.*

If we output a hypothesis in $B_\ell$, clearly, the prompting hypotheses had failed to empty out the bucket. Therefore, the last time the algorithm was iterating over all $i$'s in $[n]$, the IS-PROMPTING procedure returned `false` for every single $H_i$, and no "restart" happened. However, during those iterations, at some point, $i$ reached $i^*$, implying that the IS-PROMPTING procedure did not catch that $H_{i^*}$ is a prompting hypothesis. This event happens with probability at most $\delta_{fn}$.

$\mathbf{Pr}[\mathcal{E}_\ell] \leq \mathbf{Pr}[\text{IS-PROMPTING returns } \texttt{false} \text{ for } H_{i^*} \text{ and } H_{i^*} \text{ is a prompting hypothesis}]$

$\qquad \leq \mathbf{Pr}[\text{IS-PROMPTING returns } \texttt{false} \text{ for } H_{i^*} \mid H_{i^*} \text{ is a prompting hypothesis}] \leq \delta_{fn}.$
$$\tag{5}$$

Given Equation (4) and Equation (5), we bound the probability of outputting a far $\hat{H}$ as follows:

$$\mathbf{Pr}\Big[ \text{Outputting } \hat{H} \text{ and } \|\hat{H} - P\|_{\mathrm{TV}} > 3\,\mathsf{OPT} + \epsilon \Big]$$

$$= \sum_{\ell=1}^{k} \mathbf{Pr}\Big[ \|\hat{H} - P\|_{\mathrm{TV}} > 3\,\mathsf{OPT} + \epsilon \mid \mathcal{E}_\ell \Big] \cdot \mathbf{Pr}[\mathcal{E}_\ell]$$

$$+ \sum_{\ell=1}^{k} \mathbf{Pr}\Big[ \|\hat{H} - P\|_{\mathrm{TV}} > 3\,\mathsf{OPT} + \epsilon \mid \overline{\mathcal{E}_\ell} \Big] \cdot \mathbf{Pr}\big[\overline{\mathcal{E}_\ell}\big]$$

$$\leq \sum_{\ell=1}^{k} \mathbf{Pr}[\mathcal{E}_\ell] + \sum_{\ell=1}^{k} \gamma \cdot \mathbf{Pr}\big[\overline{\mathcal{E}_\ell}\big]$$

$$\leq k \cdot \delta_{fn} + \gamma \leq 2\,\delta'.$$

---

[5]It is worth noting that we are using the $\overline{\mathcal{E}_\ell}$ notation somewhat non-rigorously here. This notation does not represent the complementary event of $\mathcal{E}_\ell$. Instead, it denotes the complementary event of $\mathcal{E}_\ell$ conditioning on the outputting of $\hat{H}$ from the bucket $B_\ell$.

The last line is due to the fact that we set $\delta_{fn} = \delta'/k$ and $\gamma = \delta'$.

**Bounding the overall error probability:** We have shown that if we output $\hat{H}$, it is an acceptable answer with high probability. Now, we need to show that the overall error probability of the algorithm is not too high as well.

Our first claim is that the algorithm always ends, meaning that we do not restart the "for" loop infinitely many times in Line 24. Note that the restart happens only after we check that at least $\gamma/2$ of the hypotheses are removed from $B_\ell$ in Line 22. Therefore, the total number of restarts for any given bucket is at most $\log_{1/(1-\gamma/2)}(|B_\ell|) + 1 \le 4\log(n)/\gamma$ times.

In addition, we claim the algorithm does not end without outputting $\hat{H}$ or $\perp$. Every $H_i$ belongs to some bucket at every step of the algorithm. This is due to the assumption that $\hat{w}_j(H_i)$ is in $[0, 1]$. Thus, $\tilde{W}(H_i)$ is always between $0 \le \tilde{W}(H_i) \le 1 < k\epsilon'$. Thus, it is not possible to reach $|B_\ell| = 0$ for all $\ell \in [k]$ in Line 25. For the bucket that cannot be emptied out, the algorithm outputs either $\perp$ or $\hat{H}$.

Next, we show that the probability of outputting $\perp$ is at most $\delta'$. Note that the algorithm outputs $\perp$ when $H_i$ was not able to remove at least $\gamma/2$ of the hypotheses in $B_\ell$. As we have established earlier, the restart happens only $4\log(n)/\gamma$ times for every bucket. Therefore, the IS-PROMPTING procedure is invoked at most $4kn\log(n)/\gamma$ times overall. Each time we have a chance of $\delta_{fp}$ of declaring a hypothesis prompting erroneously. The algorithm in Line 22 checks if the error has happened, and it outputs $\perp$ when it does. Hence, we have:

$$\mathbf{Pr}[\text{Outputting } \perp] \le \frac{4kn\log(n)}{\gamma} \cdot \delta_{fp} \le \delta'. \tag{6}$$

The final inequality follows from setting $\delta_{fp} = \gamma \cdot \delta'/(4\,k\,n\,\log(n))$. Thus, we can bound the overall error probability as follows:

$$\mathbf{Pr}[\text{Wrong answer}] = \mathbf{Pr}[\text{Outputting } \perp]$$
$$+ \mathbf{Pr}\left[\text{Outputting } \hat{H} \text{ s.t. } \|\hat{H} - P\|_{\mathrm{TV}} > 3\,\mathsf{OPT} + \epsilon\right] \le 3\,\delta' \le \delta.$$

The last inequality holds since $\delta' = \delta/3$.

**Running time:** The initialization takes $O(n+k)$ time. The algorithm runs over $k$ buckets. For each bucket, we may restart the "for" loop $O(\log(n)/\gamma)$ times. Within each "for" loop before restarting, we invoke the IS-PROMPTING procedure $O(n)$ times where, with the exception of one of them, all return `false`. Let $T_{\texttt{false}}^{(i)}$ indicate the time that the procedure spends on $H_i$ in the `false` case. Also, set this quantity be $T_{\texttt{true}}$ if the procedure returns `true` (which happens at most once before restarting). If we find a prompting hypothesis, we query the $\hat{w}_i(H_j)$'s $O(n)$ times, and spend $O(n \cdot T_q)$ for this step. Hence, the total time complexity is:

$$\text{Total time complexity } = O\left(k \cdot \frac{\log(n)}{\gamma} \cdot \left(\sum_i^n T_{\texttt{false}}^{(i)} + T_{\texttt{true}} + n \cdot T_q\right)\right).$$

**Advanced version of IS-PROMPTING:** We described the guarantees of the advanced version of the IS-PROMPTING procedure in Lemma A.3. In that lemma, we have shown that the running time of the `false` case is proportionate to the number of hypotheses we remove from $B_\ell$ during the procedure of IS-PROMPTING. We denote this quantity in the $r$-th search for the prompting hypothesis $H_i$ in bucket $b_\ell$ via $n_r^{(\ell,i)}$. Since the total number of hypotheses that we can remove is at most $O(n)$, we have:

$$\sum_i^n T_{\texttt{false}}^{(i)} = \sum_i^n O\left(\left(\log\left(\delta_{fn}^{-1}\right) + \log\log\left(\delta_{fp}^{-1}\right)\right) \cdot T_q/\gamma^2 + n_r^{(\ell,i)} \cdot T_q/\gamma\right)$$

$$= O\left(n \cdot \left(\log\left(\delta_{fn}^{-1}\right) + \log\log\left(\delta_{fp}^{-1}\right)\right) \cdot T_q/\gamma^2 + n \cdot T_q/\gamma\right)$$

$$= O\left(n \cdot \left(\log\left(\delta_{fn}^{-1}\right) + \log\log\left(\delta_{fp}^{-1}\right)\right) \cdot T_q/\gamma^2\right)$$

Using the setting of our parameters, $\delta_{fp} = \gamma \cdot \delta'/(4\,k\,n\,\log(n)) = O(n\log n/(\delta\,\epsilon))$ and $\delta_{fn} = \delta'/k = O(1/(\delta\,\epsilon))$.

$$\sum_i^n T_{\texttt{false}}^{(i)} = O\left(\frac{n}{\delta^2} \cdot \left(\log\left(\frac{1}{\delta\,\epsilon}\right) + \log\log\left(\frac{n}{\delta\,\epsilon}\right)\right) \cdot T_q\right)$$

$$= O\left(\frac{n}{\delta^2} \cdot \left(\log\left(\frac{1}{\delta\,\epsilon}\right) + \log\log\left(n\right)\right) \cdot T_q\right).$$

Moreover, for the case that a prompting hypothesis is found:

$$T_{\texttt{true}} = O\left(\log\left(\delta_{fp}^{-1}\right) \cdot \left(\log\left(\delta_{fn}^{-1}\right) + \log\log\left(\delta_{fp}^{-1}\right)\right) \cdot T_q/\gamma^2\right)$$

$$= O\left(\frac{1}{\delta^2} \cdot \log\left(\frac{n}{\delta\,\epsilon}\right) \cdot \left(\log\left(\frac{1}{\delta\,\epsilon}\right) + \log\log\left(\frac{n}{\delta\,\epsilon}\right)\right) \cdot T_q\right)$$

$$= O\left(\frac{1}{\delta^2} \cdot \log\left(\frac{n}{\delta\,\epsilon}\right) \cdot \left(\log\left(\frac{1}{\delta\,\epsilon}\right) + \log\log\left(n\right)\right) \cdot T_q\right).$$

Hence, the total time complexity is:

$$\text{Total time complexity} = O\left(k \cdot \frac{\log(n)}{\gamma} \cdot \left(\sum_i^n T_{\texttt{false}}^{(i)} + T_{\texttt{true}} + n \cdot T_q\right)\right)$$

$$= O\left(\frac{\log(n)}{\delta^3\,\epsilon} \cdot \left(n \cdot \log\log n + n \cdot \log\left(\frac{1}{\delta\,\epsilon}\right) + \left(\log\left(\frac{1}{\delta\,\epsilon}\right)\right)^2\right) \cdot T_q\right)$$

$$= \tilde{O}\left(\frac{n}{\delta^3\epsilon} \cdot T_q\right).$$

**Simple version of Is-Prompting:** In this version, we only have one possible confidence parameter, which is $\min(\delta_{fn}, \delta_{fp})$. Hence, using Lemma A.2, we have:

$$T_{\texttt{false}}^{(i)} = T_{\texttt{true}} = O\left(\log\left((\min(\delta_{fn}, \delta_{fp}))^{-1}\right) \cdot T_q/\gamma^2\right).$$

$$\text{Total time complexity} = O\left(k \cdot \frac{\log(n)}{\gamma} \cdot \left(\sum_i^n T_{\texttt{false}}^{(i)} + T_{\texttt{true}} + n \cdot T_q\right)\right)$$

$$= O\left(n \cdot \frac{\log(n)}{\delta^3\,\epsilon} \cdot \left(\log\left(\frac{n}{\delta\,\epsilon}\right)\right) \cdot T_q\right)$$

$$= \tilde{O}\left(\frac{n}{\delta^3\epsilon} \cdot T_q\right).$$

Thus, the proof is complete. $\qquad\square$

## A.1 Identifying prompting hypothesis

In this section, we provide two algorithms for identifying a prompting hypothesis. The simple algorithm, presented in Section A.1.1, involves sampling the hypotheses in a bucket to estimate the fraction of the hypotheses that $H_i$ can move, using a standard application of the Hoeffding bound. The advanced algorithm, presented in Section A.1.2, is based on the fact that we require different confidence parameters for false negative and false positive errors, given our analysis for Algorithm 1. We also show that the time complexity of this algorithm, in the case it returns `false`, is proportional to the number of hypotheses we can remove from the bucket during this procedure, which later helps us to use an amortized argument for the overall time spent on `false` returns.

### A.1.1 Simple version of IS-PROMPTING

---

**Algorithm 2** An algorithm to identify a prompting hypothesis (simple version).

1: **procedure** IS-PROMPTING-SIMPLE($\mathcal{B}$, $\ell$, $H_i$, $\gamma$, $\epsilon'$, $\delta$, and query access to $\hat{w}_j(H_i)$'s)
2:     $t \leftarrow \lceil 8 \log(2/\delta)/\gamma^2 \rceil$
3:     $u_r \leftarrow 0$                     ▷ $u_r$ counts # sampled hypotheses that $H_i$ updates
4:     **repeat** $t$ **times**
5:         $H_j \leftarrow$ a random hypothesis in $B_\ell$
6:         **if** $\hat{w}_i(H_j) > \tilde{W}(H_j) + \epsilon'$ **then**
7:             $u_r \leftarrow u_r + 1$
8:
9:     **if** $\frac{u_r}{t} < \frac{3\gamma}{4}$ **then**
10:         **return** `false`
11:     **else**
12:         **return** `true`

---

**Lemma A.2.** *Suppose we are given two parameters $\delta, \epsilon' \in (0,1)$, and three positive integers $n$, $k$, and $\ell \in [k]$. Assume we are given a class of hypotheses $\mathcal{H}$, its partition into $k$ buckets $\mathcal{B} = \{B_1, B_2, \ldots, B_k\}$, and a hypothesis $H_i \in \mathcal{H}$. The IS-PROMPTING-SIMPLE procedure in Algorithm 2 receives $\mathcal{H}, \mathcal{B}, \ell, H_i, \delta_{fp}, \delta_{fn}$, and $\epsilon'$ as its input and returns `true` or `false` with the following guarantees:*

- *If $H_i$ substantially updates at least a $\gamma$-fraction of the hypotheses in $B_\ell$, then IS-PROMPTING-SIMPLE returns `true` with probability at least $1 - \delta$.*

- *If $H_i$ substantially updates less than a $\gamma/2$-fraction of the hypotheses in $B_\ell$, then IS-PROMPTING-SIMPLE returns `false` with probability at least $1 - \delta$.*

- *We have the following guarantees for the algorithm:*

  *Number of queries to $\hat{w}_i(H_j)$'s: $O\left(\log(\delta^{-1})/\gamma^2\right)$ ,*

  *Running time: $O\left(\log(\delta^{-1}) \cdot T_q/\gamma^2\right)$ ,*

  *where $T_q$ is the time to obtain each $\hat{w}_j(H_i)$.*

*Proof.* This is a simple application of the Hoeffding bound:

$$\mathbf{Pr}\left[\left|\frac{u}{t} - \mathbf{E}\left[\frac{u}{t}\right]\right| \geq \frac{\gamma}{4}\right] \leq 2\exp\left(-2t\gamma^2\right) \leq \delta.$$

$\square$

### A.1.2 Advanced version of IS-PROMPTING

We present an overview of the algorithm for the advanced version of IS-PROMPTING. Suppose we are given a parameter $\gamma$, and our goal is to design an algorithm that does the following with high probability: it returns `true` when $H_i$ can substantially update a $\gamma$-fraction of the hypotheses in $B_\ell$, and it returns `false` when $H_i$ can substantially update less than a $\gamma/2$ fraction of the hypotheses in $B_\ell$.

For this new procedure, we take advantage of an asymmetry that exists between returning `false` and `true`. This asymmetry arises mainly from the analysis of Algorithm 1. Roughly speaking, for every bucket $\ell$, to find a single prompting hypothesis, we check $O(n)$ (potentially non-prompting) hypotheses. On the other hand, we only need $\log_{1/\gamma}(B_\ell)$ prompting hypotheses to empty out the whole bucket. Hence, we expect that the procedure returns mostly `false` rather than `true`.

Another difference here is the cost we pay for mistakenly declaring `true` or `false`. When we declare a hypothesis prompting, we iterate over all hypotheses in $B_\ell$ and check if we can remove them from the bucket. If the hypothesis was not actually prompting, we have paid a substantial cost (in time) without making progress. Thus, we cannot have too many hypotheses that the algorithm declares prompting while they are not (false positives). However, the probability of mistakenly missing a prompting hypothesis does not have to be this small (false negatives).

Below is a more precise definition of these errors:

$$\delta_{fp} := \mathbf{Pr}[\text{Returning } \texttt{true} \mid H_i \text{ substantially updates less than a } \gamma/2 \text{ fraction}]$$
$$\delta_{fn} := \mathbf{Pr}[\text{Returning } \texttt{false} \mid H_i \text{ substantially updates at least a } \gamma \text{ fraction}]$$

Given the analysis of our algorithm, we can show that the probability of false positives must be around $\tilde{O}(1/|B_\ell|) = \tilde{O}(1/n)$. On the other hand, our analysis is robust as long as the probability of false negatives is roughly $O(\delta)$, where $\delta$ is the overall confidence of our algorithm. (The reason for this choice of parameter may not be obvious from this high-level discussion; however, it is a direct artifact of our analysis.) This gap is particularly large when $\delta$ is a small constant, as is common in the literature.

The IS-PROMPTING procedure (Algorithm 3) is designed in a way that has different time complexities depending on its output being `true` or `false`. The procedure uses more than $O(\log n)$ (more like $O(\log n \cdot \log(1/\delta)/\gamma^2)$) time if it returns `true`. However, the main advantage is that it uses much less time when the output is `false`. In fact, instead of paying $O(\log n)$, we only spend $O(\log(1/\delta))$ time plus an amortized cost of $O(1/\gamma)$ for the $O(B_\ell)$ calls to the procedure.

The algorithm runs in roughly $O(\log n)$ rounds. In each round, it draws a few random hypotheses from $B_\ell$ (roughly $O(\log(1/\delta)/\gamma^2)$). In each round, we check if the fraction of hypotheses $H_i$ can substantially update is close to $\gamma$. At any round, if we see that the fraction is not close to $\gamma$, we return `false`. The hope is that for a non-prompting hypothesis, we either stop very quickly or the hypothesis passes too many rounds without us noticing that it is not prompting. Therefore, we must have seen an inflated number of substantial updates in these rounds. Before returning `false`, we perform those substantial updates among the sampled hypotheses that we have observed to remove those movable hypotheses from $B_\ell$.

We can capitalize on this fact in our cost analysis. In fact, we can show that in the `false` case, if the procedure takes roughly $O(t/\gamma)$ time, it must have removed $t$ hypotheses from the bucket. Thus, one can show that the amortized cost of these elongated rounds is only $O(1/\gamma)$.[6]

**Lemma A.3.** *Suppose we are given three parameters $\delta_{fp}, \delta_{fn}, \epsilon' \in (0,1)$, and three positive integers $n$, $k$, and $\ell \in [k]$. Assume we are given a classes of hypotheses $\mathcal{H}$, its partition into $k$ buckets $\mathcal{B} = \{B_1, B_2, \ldots, B_k\}$ and a hypothesis $H_i \in \mathcal{H}$. The IS-PROMPTING procedure in Algorithm 3 receives $\mathcal{H}, \mathcal{B}, \ell, H_i, \delta_{fp}, \delta_{fn}$, and $\epsilon'$ as its input and returns `true` or `false` with the following guarantees:*

- *If $H_i$ substantially updates at least $\gamma$ fraction of the hypotheses in $B_\ell$, then IS-PROMPTING returns `true` with probability at least $1 - \delta_{fn}$.*

- *If $H_i$ substantially updates less than $\gamma/2$ fraction of the hypotheses in $B_\ell$, then IS-PROMPTING returns `false` with probability at least $1 - \delta_{fp}$.*

---

[6]The parameters in this high-level discussion lack precision. For a rigorous analysis, refer to our proof of Lemma A.3.

---

**Algorithm 3** An algorithm to identify a prompting hypothesis.

---

1: **procedure** IS-PROMPTING($\mathcal{B}$, $\ell$, $H_i$, $\gamma$, $\epsilon'$, $\delta_{fn}$, $\delta_{fp}$, and query access to $\hat{w}_j(H_i)$'s)

2: $\quad R \leftarrow \left\lceil \log_{3/2}\left(\frac{1}{\delta_{fp}}\right)\right\rceil$

3: $\quad S_0 \leftarrow \emptyset$ $\qquad\qquad\qquad\qquad$ ▷ $S$ is a set of hypotheses that $H_i$ can update substantially.

4: $\quad$ **for** $r = 1, \ldots, R$ **do**

5: $\qquad$ **if** $\frac{|S_{r-1}|}{B_\ell} \geq \frac{\gamma}{2}$ **then**

6: $\qquad\qquad$ **return** true

7: $\qquad$ $S_r \leftarrow S_{r-1}$

8: $\qquad$ $u_r \leftarrow 0$ $\qquad\qquad\qquad\qquad$ ▷ $u_r$ indicates # hypotheses $H_i$ updates

9: $\qquad$ **repeat** $t := \left\lceil \frac{48\log(2R/\delta_{fn})}{\gamma^2}\right\rceil$ **times**

10: $\qquad\qquad$ $G_j^{(r)} \leftarrow$ a random hypothesis in $B_\ell$

11: $\qquad\qquad$ **if** $\hat{w}_i\left(G_j^{(r)}\right) > \tilde{W}\left(G_j^{(r)}\right) + \epsilon'$ **then**

12: $\qquad\qquad\qquad$ $u_r \leftarrow u_r + 1$

13: $\qquad\qquad\qquad$ $S_r \leftarrow S_r \cup \{G_j^{(r)}\}$

14:

15: $\qquad$ **if** $|S_r| - |S_{r-1}| < \frac{\gamma \cdot t}{8}$ **then**

16: $\qquad\qquad$ **for** $H_j \in S_r$ **do**

17: $\qquad\qquad\qquad$ $\tilde{W}(H_j) \leftarrow \max\left(\tilde{W}(H_j), \hat{w}_i(H_j)\right)$

18: $\qquad\qquad\qquad$ Move $H_j$ to $B_{\lceil \tilde{W}(H_j)/\epsilon'\rceil}$

19: $\qquad\qquad$ **return** false

20: $\qquad$ **if** $\frac{u_r}{t} < \frac{3\gamma}{4}$ **then**

21: $\qquad\qquad$ **for** $H_j \in S_r$ **do**

22: $\qquad\qquad\qquad$ $\tilde{W}(H_j) \leftarrow \max\left(\tilde{W}(H_j), \hat{w}_i(H_j)\right)$

23: $\qquad\qquad\qquad$ Move $H_j$ to $B_{\lceil \tilde{W}(H_j)/\epsilon'\rceil}$

24: $\qquad\qquad$ **return** false

25: $\quad$ **return** true

---

- *If the algorithm returns* true, *we have the following guarantees for the algorithm:*

  *Number of queries to $\hat{w}_i(H_j)$'s:* $O\left(\log\left(\delta_{fp}^{-1}\right) \cdot \left(\log\left(\delta_{fn}^{-1}\right) + \log\log\left(\delta_{fp}^{-1}\right)\right)/\gamma^2\right)$,

  *Running time:* $O\left(\log\left(\delta_{fp}^{-1}\right) \cdot \left(\log\left(\delta_{fn}^{-1}\right) + \log\log\left(\delta_{fp}^{-1}\right)\right) \cdot T_q/\gamma^2\right)$.

- *Suppose the algorithm returns* false. *Assume $n_r^{(\ell,i)}$ indicates the number of hypotheses that the algorithm removes from $B_\ell$. Then, we have the following guarantees for the algorithm:*

  *Number of queries to $\hat{w}_i(H_j)$'s:* $O\left(\left(\log\left(\delta_{fn}^{-1}\right) + \log\log\left(\delta_{fp}^{-1}\right)\right)/\gamma^2 + n_r^{(\ell,i)}/\gamma\right)$,

  *Running time:* $O\left(\left(\log\left(\delta_{fn}^{-1}\right) + \log\log\left(\delta_{fp}^{-1}\right)\right) \cdot T_q/\gamma^2 + n_r^{(\ell,i)} \cdot T_q/\gamma\right)$ .$f$

*In above bounds, where $T_q$ denotes the time complexity to obtain each $\hat{w}_j(H_i)$.*

*Proof.* The probabilities in this proof are taken over the random choices of $G_j^{(r)}$'s. Note that $S$ contains at the hypotheses for which we can change.

**Probability of false positive:** First, we show that probability of outputting true, while $H_i$ can substantially update less $\gamma/2$ fraction of hypotheses in $B_\ell$, is at most $\delta_{fp}$. The algorithm return true in two places. First, at the beginning of each round, we check whether $S_{r-1}$ contains at least $\gamma/2$-fraction of hypotheses. This answer is always correct with probability one. Since the algorithm

has an evidence that $H_i$ can certainly update at least $\gamma/2$ of hypotheses in $B_\ell$. Thus, this case does not affect our false positive rate $\delta_{fp}$.

Second, the algorithm returns `true` after all rounds end. This case only happens when $u_r/t$ is at least $3\gamma/4$ in every round in Line 20. It is not hard to see that $u_r$ is a binomial random variable with $t$ trials. The success probability of each trial is the ratio of the hypotheses in $B_\ell$ that $H_i$ can update substantially. Thus, in the case where $H_i$ substantially updates less than $\gamma$ fraction of hypotheses, by Markov's inequality, we have:

$$\mathbf{Pr}\big[\text{Returning } \texttt{true} \mid H_i \text{ substantially updates less than } \gamma/2\text{-fraction}\big]$$

$$= \mathbf{Pr}\left[\forall r \in [R] : \frac{u_r}{t} \geq \frac{3\gamma}{4}\right] \left(\mathbf{Pr}\left[\frac{u_r}{t} \geq \frac{3\gamma}{4}\right]\right)^R \leq (2/3)^R \leq \delta_{fp}$$

**Probability of false negative:** Suppose that $H_i$ substantially updates at least $\gamma$-fraction of the hypotheses in $B_\ell$. That is, $H_i$ can update each $G_j^{(r)}$ with probability at least $\gamma$. Fix a round $r$. We bound the probability of returning `false` in Line 19 and Line 24.

In Line 19, we return `false` when $|S_r| - |S_{r-1}| < \frac{\gamma \cdot t}{8}$. That is, the number of new hypotheses we added to $S_r$ is not too large. Consider the random hypotheses we draw at round $r$. Our claim is that most of these random hypotheses are not in $S_{r-1}$. The probability that one $G_j^{(r)}$ be selected from $S_{r-1}$ is $|S_{r-1}| / |B_\ell|$. However, this expectation is less than $\gamma/2$. Otherwise, the algorithm would have returned `true` earlier in Line 5. With this information in mind, we can use the Hoeffding bound and get:

$$\mathbf{Pr}\left[\text{number of } G_j^{(r)} \in S_{r-1} \geq \frac{5\,\gamma \cdot t}{8}\right] = \mathbf{Pr}\left[\frac{\text{number of } G_j^{(r)} \in S_{r-1}}{t} \geq \frac{5\,\gamma}{8} = \frac{\gamma}{2} \cdot \left(1 + \frac{1}{4}\right)\right]$$

$$\leq \mathbf{Pr}\left[\frac{\text{number of } G_j^{(r)} \in S_{r-1}}{t} > \frac{|S_{r-1}|}{|B_\ell|} + \frac{\gamma}{4}\right]$$

$$\leq \exp\left(-2\,t\,(\gamma/4)^2\right) \leq \frac{\delta_{fn}}{2\,R}.$$

In Line 24, we return `false` if $u_r/t$ is less than $3\,\gamma/4$. By Chernoff bound, we have:

$$\mathbf{Pr}\left[\frac{u_r}{t} < \gamma \cdot \left(1 - \frac{1}{4}\right)\right] \leq \exp\left(-\frac{t\,\gamma}{3 \cdot 4^2}\right) = \frac{\delta_{fn}}{2\,R}.$$

Taking the union bound over all rounds, we obtain:

$$\mathbf{Pr}\big[\text{Returning } \texttt{false} \mid H_i \text{ substantially updates at least } \gamma\text{-fraction}\big] \leq \delta_{fn}.$$

**Running time:** In the case the algorithm return `true`, the algorithm makes $O(R \cdot t)$ queries to $\hat{w}_i\left(G_j^{(r)}\right)$'s. And, it runs in the $O(R \cdot t \cdot T_q)$. If algorithm returns `false` in round $r$, similarly it makes $O(r \cdot t)$ queries to $\hat{w}_i\left(G_j^{(r)}\right)$'s. And, it runs in the $O(r \cdot t \cdot T_q)$. Note that with the exception of the last round in every round $r' \in [r-1]$. At least $(\gamma \cdot t)/8$ new hypotheses are added to $S_{r'}$. Therefore, the size of $S_r$ is at least $(\gamma \cdot t \cdot (r-1))/8$. We remove all of these hypotheses before we output false which cause the size of $B_\ell$ to drop by $n_r^{(\ell,i)} := |S_r|$. In this case, the algorithm makes $O(t + n_r^{(\ell,i)}/\gamma)$ queries to $\hat{w}_i\left(G_j^{(r)}\right)$'s. And, it runs in the $O\left(\left(t + n_r^{(\ell,i)}/\gamma\right) \cdot T_q\right)$. Thus, the proof is complete by setting $R := \left\lceil \log_{3/2}(1/\delta_{fp})\right\rceil$ and $t := \left\lceil 48 \log(2R/\delta_{fn})/\gamma^2\right\rceil$. $\qquad\square$

## A.2 Proof of Lemma A.1

**Lemma A.1.** *Suppose $B_\ell \subseteq \mathcal{H}$ is the smallest non-empty bucket for which at least a $\gamma$ fraction of its hypotheses are $(3\,\mathsf{OPT} + \epsilon)$-far from $P$. Then $H_{i^*}$ is a prompting hypothesis that can remove a $\gamma$ fraction of the hypotheses in $B_\ell$.*

*Proof.* We show that $H_{i^*}$ can substantially update $\tilde{W}$ of any far hypothesis in $B_\ell$. Let $H_f$ denote a $(3\,\mathsf{OPT} + \epsilon)$-far hypothesis. First, observe that $\hat{w}_{i^*}(H_f)$ must be large compared to $\mathsf{OPT}$:

$$
\begin{aligned}
\hat{w}_{i^*}(H_f) &\geq w_{i^*}(H_f) - \epsilon' = |H_f(\mathcal{S}_{f\,i^*}) - P(\mathcal{S}_{f\,i^*})| - \epsilon' \\
&\geq |H_f(\mathcal{S}_{f\,i^*}) - H_{i^*}(\mathcal{S}_{f\,i^*})| - |H_{i^*}(\mathcal{S}_{f\,i^*}) - P(\mathcal{S}_{f\,i^*})| - \epsilon' \\
&\geq \|H_f - H_{i^*}\|_{\mathrm{TV}} - \|H_{i^*} - P\|_{\mathrm{TV}} - \epsilon' \qquad \text{(By Fact C.2)} \\
&\geq \|H_f - P\|_{\mathrm{TV}} - 2\,\|H_{i^*} - P\|_{\mathrm{TV}} - \epsilon' \\
&> 3\,\mathsf{OPT} + \epsilon - 2\,\mathsf{OPT} - \epsilon' \geq \mathsf{OPT} + 2\,\epsilon'.
\end{aligned}
$$

In addition, $\mathsf{OPT}$ cannot be much smaller than $\ell\,\epsilon'$. The main reason is that all the buckets before $B_\ell$ are empty. Hence, $H_{i^*}$ belongs to a bucket $B_{\ell'}$ where $\ell'$ is at least $\ell$. By our definition of bucketing: $\tilde{W}(H_{i^*})$ must be at least $(\ell - 1)\,\epsilon'$. Hence we have:

$$
(\ell - 1)\,\epsilon' \leq \tilde{W}(H_{i^*}) \leq \hat{W}(H_{i^*}) \leq W(H_{i^*}) \leq \mathsf{OPT}.
$$

Note that we assume $H_f$ belongs to $B_\ell$, so $\tilde{W}_f$ is less than $\ell\epsilon'$. Putting all these observation together, we achieve:

$$
\tilde{W}_f + \epsilon' < \ell\,\epsilon' + \epsilon' \leq \mathsf{OPT} + 2\epsilon' < \hat{w}_{i^*}(H_f).
$$

Therefore, if we update $\tilde{W}_f$ via $\hat{w}_{i^*}(H_f)$, then $H_{i^*}$ can significantly improve $\tilde{W}_f$. This fact implies that $H_{i^*}$ can removes all the bad hypothesis from $B_\ell$ which concludes the proof of the lemma. $\square$

# B  Linear time algorithm with $\alpha = 4$

In this section, we present our second algorithm whose sample complexity has a better dependency on the accuracy parameter $\epsilon$. An overview of our approach is presented in Section 3.3. The pseudocode of our approach is presented in Algorithm 4. We prove the performance of this algorithm in Theorem 6.

Apart from improving the accuracy parameters, we also provide modifications to our algorithm that can achieve low rounds of adaptivity (roughly speaking, it means the number of times the algorithm needs to look at the sample set). More formally, in our access model, we define *one round of adaptivity* as follows: The algorithm selects a set of $t = O(n^2)$ pairs of indices $\{(i_\ell, j_\ell)\}_{\ell \in [t]}$ and query all $\hat{w}_{j_\ell}(H_{i_\ell})$ at once. The number of adaptivity rounds represents the total count of times the algorithm needs to repeat this process in order to produce its final output. This is particularly interesting when considering a federated learning setting in which the rounds of interactivity are important. The modification of our algorithm runs in time $O(n^{1+\lambda})$ runs in $O(1/\lambda)$ rounds of adaptivity.

**Theorem 6.** *Suppose Algorithm 4 receives these parameters as input: $\sigma \in [0,1]$, $\eta \in (0, 1/4)$, $\epsilon, \delta \in (0,1)$. Also, assume the algorithm has access to a class of $n$ hypotheses $\mathcal{H}$, and a set of unmarked hypotheses in $\mathcal{Q} \subseteq \mathcal{H}$ where initially $\mathcal{Q} = \mathcal{H}$. Suppose the algorithm can query $\hat{w}_j(H_i)$'s with error at most $\epsilon' \leq \epsilon/6$. The algorithm has one of the two possible outcomes: either it declares that $\sigma < \mathsf{OPT}$; or it outputs a hypothesis $H_i$ such that: with probability at least $1 - \delta$ we have:*

$$
\|\hat{H} - P\|_{TV} \leq 4\,\max(\sigma, \mathsf{OPT}) + \epsilon.
$$

*The algorithm runs in $O(n/\eta \cdot T_q)$ time and $O\left(\log^2_{1/\eta}(n)\right)$ rounds of adaptivity. Here $T_q$ is the time required to obtain each $\hat{w}_j(H_i)$.*

---

**Algorithm 4** finds a hypothesis that is $4 \max(\sigma, \mathsf{OPT}) + \epsilon$ close to $P$.

---

1: **procedure** SELECT-HYPOTHESIS($\mathcal{H}$, $\mathcal{Q}$, $\sigma$, $\epsilon$, $\delta$, $\eta$, , and query access to $\hat{w}_j(H_i)$'s)
2:     **if** $|\mathcal{Q}| = 0$ **then**
3:         **return** "$\sigma < \mathsf{OPT}$".
4:     **if** $|\mathcal{Q}| = 1$ **then**
5:         $H_i \leftarrow$ the only unmarked hypothesis left.
6:         **if** $\hat{W}(H_i) > \sigma + \epsilon'$ **then**
7:             **return** "$\sigma < \mathsf{OPT}$".
8:         **else**
9:             **return** $H_i$.
10:     $\epsilon' \leftarrow \epsilon/6$
11:     $\delta'' \leftarrow \delta / \left( 100 \cdot \left( \log_{1/(2\eta)}(n) \cdot \log_{1/(\eta)}(n) \right) \right)$
12:     $m \leftarrow \lfloor \eta \cdot n \rfloor$
13:     Run FIND-SEED($\mathcal{H}$, $\mathcal{Q}$, $\sigma$, $\epsilon'$, $\delta''$, $\eta$)
14:     **if** FIND-SEED declares "$\sigma < \mathsf{OPT}$" **then**
15:         **return** "$\sigma < \mathsf{OPT}$".
16:     **else if** FIND-SEED returns "start over" **then**
17:         **return** SELECT-HYPOTHESIS($\mathcal{H}$, $\mathcal{Q}$, $\sigma$, $\epsilon$, $\delta$, $\eta$)
18:     $H_i \leftarrow$ the seed that FIND-SEED found.
19:     **while** $m > \frac{1}{\eta}$ **do**
20:         Run BOOST-SEED($\mathcal{H}$, $\mathcal{Q}$, $H_i$, $\sigma$, $\kappa = 2$, $\kappa' = 2$, $\epsilon'$, $\eta$)
21:         **if** BOOST-SEED finds the final answer. **then**
22:             **return** $H_i$ found by BOOST-SEED.
23:         **else if** BOOST-SEED finds a new seed **then**
24:             $H_i \leftarrow$ the seed that BOOST-SEED returns
25:             $m \leftarrow \lfloor \eta \cdot m \rfloor$
26:         **else if** BOOST-SEED returns "start over" **then**
27:             **return** SELECT-HYPOTHESIS($\mathcal{H}$, $\mathcal{Q}$, $s$, $\sigma$, $\epsilon$, $\delta$)
28:     $H_\ell \leftarrow \arg\min_{H_j \in \mathcal{M}} \hat{W}(H_j)$             ▷ $\mathcal{M}$ denotes $\mathcal{M}_{2\sigma + \epsilon', \, 4\sigma + 5\epsilon'}(H_i)$.
29:     **if** $\hat{W}(H_\ell) \le 2\sigma + \epsilon'$ **then**
30:         Output $H_\ell$.
31:     **else**
32:         Output $H_i$.

---

*Proof.* For the sake of argument, assume each of the sub-routines in the algorithm works as guaranteed with probability one (as oppose to the case where they work as expected with probability $1 - \delta''$). Later on, we discuss the overall confidence of the algorithm to remove this assumption.

**Accuracy of seeds:** the procedure to find the initial seeds provides us with a $(\sigma + \epsilon', 3\sigma + 3\epsilon', m)$-seed with high probability. Using Fact C.4, this seed is also a $(2\sigma + \epsilon', 4\sigma + 5\epsilon', m)$. In boosting seeds, we start with $(2\sigma + \epsilon', 4\sigma + 5\epsilon', m)$ and we get $(2\sigma + \epsilon', 4\sigma + 5\epsilon', m')$ where $m' = \lfloor \eta \cdot m \rfloor$. This statement is justified by Theorem 9, setting $\kappa = \kappa' = 2$, and using the same values for $a = 2, \sigma + \epsilon'$ and $b = 4, \sigma + 5\epsilon'$ for our seeds. It is worth noting that the only parameter that changes while we boost is $m$.

**Accuracy of the output** Consider the case where we produce output when $|\mathcal{Q}|$ is zero. That implies that our algorithm marked all the hypotheses. Throughout the course of algorithms, we never mark a hypothesis $H_i$ unless we have found an evidence that $\|H_i - P\|_{\mathrm{TV}} > \sigma$. This was established by observing $\hat{w}_j(H_i) > \sigma + \epsilon'$, which implies $\|H_i - P\|_{\mathrm{TV}}$ must be greater than $\sigma$; Or, showing a close-by hypotheses to $H_i$ is far from $P$ and apply triangle inequality. See our argument for Case 3.2 in the proof of Theorem 9 for example. Thus, if all hypotheses are marked, it must be the case that for every $H_i$, $\|H_i - P\|_{\mathrm{TV}}$ is greater than $\sigma$. Hence, when the size of $\mathcal{Q}$ is zero we can truly assert that "$\sigma < \mathsf{OPT}$".

For the case that $|\mathcal{Q}| = 1$, the algorithm focuses on $H_i$, the only unmarked hypothesis left. If we found $\hat{W}(H_i)$ is greater than $\sigma + \epsilon'$, we can simply conclude that for every $H_i$, $\|H_i - P\|_{\text{TV}}$ is greater than $\sigma$, and "$\sigma < \text{OPT}$". Otherwise, using Fact C.3, the $H_i$ is a valid answer:

$$\|H_i - P\|_{\text{TV}} \leq W(H_i) + 2\text{OPT} \leq \hat{W}(H_i) + \epsilon' + 2\text{OPT} \leq 3 \max(\sigma, \text{OPT}) + 2\epsilon'.$$

In addition, the accuracy of the $H_i$ that is returned in Line 22 is guaranteed by the Theorem 9, and setting $\kappa = \kappa' = 2$. Next, lets focus on the hypotheses we output after the "if" condition in Line 29. When $\hat{W}(H_\ell)$ is small, we have:

$$\|H_\ell - P\|_{\text{TV}} \leq W(H_\ell) + 2\text{OPT} \leq \hat{W}(H_\ell) + \epsilon' + 2\text{OPT} \leq 2\sigma + 2\epsilon' + 2\text{OPT} \leq 4 \max(\sigma, \text{OPT}) + 2\epsilon'.$$

When $\hat{W}(H_\ell)$ is large, the analysis is very similar to Case 1 in the proof of Theorem 9. Again, by setting $\kappa = \kappa' = 2$, and using the facts that $\hat{W}(H_\ell) - \epsilon' \geq \kappa\sigma$ and $\hat{W}(H_i) \leq (\kappa + 2)\sigma + 5\epsilon'$, we obtain:

$$\|H_i - P\|_{\text{TV}} \leq 4 \max(\sigma, \text{OPT}) + 6\epsilon'.$$

**Number of recursions:** Given Theorem 8 and Theorem 9, we only see "start over" when $(1 - 2\eta)$ fraction of unmarked hypotheses in $\mathcal{Q}$ have been marked. Since we have $n$ hypotheses, we do not start over more than $O(\log_{1/\eta}(n))$ times.

**Running time:** In the while loop, $m$ decreases with a factor of $\eta$ every time we find a new seed. Hence, the total number of iterations is bounded by $O(\log_{1/\eta}(n))$. Using Theorem 8 and Theorem 9, invoking the procedures for finding a seed and boosting a seed each takes $O(n \log(n/\delta'')/\eta \cdot T_q)$ time. Thus, the total time complexity is:

$$O\left(\frac{n}{\eta} \cdot \left(\log(n/\delta) + \log\log_{1/\eta}(n)\right) \cdot \log^2_{1/\eta}(n) \cdot T_q\right) = \tilde{O}\left(\frac{n \cdot \log(1/\delta) \cdot T_q}{\eta}\right).$$

**Overall confidence parameter:** The total number of subroutines we call here is at most $O(\log^2_{1/\eta}(n))$. Thus, by setting $\delta''$ to $O(\delta / \log^2_{1/\eta}(n))$ and using the union bound, we can show the overall error probability is bounded by $\delta$.

**Rounds of adaptivity:** Using Theorem 8 and Theorem 9, invoking the procedures for finding a seed and boosting a seed each takes $O(1)$ rounds of adaptivity. Hence, the overall rounds of adaptivity is $O\left(\log^2_{1/\eta}(n)\right)$ rounds.

$\square$

**Remark 7.** *One could argue that we have demonstrated the algorithm operates in $\tilde{O}(n/\eta \cdot T_q)$ time with a probability of $1 - \delta$. However, this fact does not inherently guarantee that the algorithm consistently maintains the desired time complexity. Fortunately, a straightforward solution exists to address this concern.*

*Our algorithm may fail for two primary reasons: either too few hypotheses were marked, or the identified seed had a larger $m$ value than anticipated. In either scenario, the algorithm can verify the occurrence of this undesirable, but improbable event. In these situations, we can output $\perp$ to indicate the algorithm's failure to produce a valid answer. With this adjustment, the time complexity remains low as desired.*

**Corollary B.1.** *Suppose $\mathcal{H}$ is a class of $n$ known hypotheses and $P$ is an unknown hypothesis. Let $\epsilon$ and $\delta$ be two parameters in $(0, 1)$. Assume the algorithm has access to accurate estimates of $\hat{w}_j(H_i)$'s that have error at most $\epsilon'$. If $3\epsilon' \leq \epsilon$, then for every $\eta \in (1/n, 1/4)$ there exists an $(\alpha = 4, \epsilon, \delta)$-proper learner for $P$ in $\mathcal{H}$. The running time of our algorithm is $\tilde{O}((n/\eta) \cdot \log(1/\delta) \cdot \log(1/\epsilon) \cdot T_q)$, and it can be implemented in $O\left(\log^2_{1/\eta}(n) \cdot \log(1/\epsilon)\right)$ rounds of adaptivity.*

*Proof.* This is a direct corollary of Theorem 6 combined with a binary search over values of $\sigma \in \{\epsilon', \epsilon'\}$ where $\epsilon' := \epsilon/100$. Note that for every $\sigma \geq \mathsf{OPT}$, the algorithm has to produce a hypothesis, since it cannot declare $\mathsf{OPT} < \sigma$ with high probability. Outputting the hypothesis associated with the smallest $\sigma$ would give us the desired guaranteed. $\square$

## B.1 Finding initial seed

In this section, we describe an algorithm that receives parameter $\sigma$ as input and finds a $(\sigma + \epsilon', \, 3\sigma + 2\epsilon', \, \lfloor \eta \cdot n \rfloor)$-seed in roughly $\tilde{O}(n/\eta \cdot T_q)$ time with high probability or declares "$\sigma < \mathsf{OPT}$".

---

**Algorithm 5** finds a $(\sigma + \epsilon', \, 3\sigma + 3\epsilon', \, m)$-seed

---

1: **procedure** FIND-SEED($\mathcal{H}, \, \mathcal{Q}, \, \sigma, \, \epsilon', \, \delta, \, \eta$, and query access to $\hat{w}_j(H_i)$'s)
2:      $m \leftarrow \lfloor \eta \cdot n \rfloor$
3:      **for** $i = 1, \ldots, n$ **do**
4:          **repeat** $t := \lceil 8\, n \log (2\, n/\delta) / m \rceil$ **times**
5:              $H_j \leftarrow$ a uniformly random sample drawn from $\mathcal{H}$
6:              **if** $\hat{w}_j(H_i) > \sigma + \epsilon'$ **then**
7:                  Mark $H_i$ in $\mathcal{Q}$.
8:              $H_j \leftarrow$ a uniformly random sample drawn from $\mathcal{Q}$
9:              **if** $\hat{w}_j(H_i) > \sigma + \epsilon'$ **then**
10:                 Mark $H_i$ in $\mathcal{Q}$.
11:
12:          **if** $|\mathcal{Q}| \leq t$ **then**
13:              **for** every $H_j \in \mathcal{Q}$ **do**
14:                 **if** $\hat{w}_j(H_i) > \sigma + \epsilon'$ **then**
15:                    Mark $H_i$ in $\mathcal{Q}$.
16:          **if** $H_i$ is unmarked. **then**
17:              Compute $\hat{W}(H_i)$.
18:              **if** $\hat{W}(H_i) \leq 3\sigma + 3\epsilon'$ **then**
19:                 **return** $H_i$ as a seed.
20:              **else**
21:                 **for** $H_j \in \mathcal{H}$ **do**
22:                    **if** $\hat{w}_i(H_j) > \sigma + \epsilon'$ or $\hat{w}_j(H_i) \leq \sigma + \epsilon'$ **then**
23:                       Mark $H_j$ in $\mathcal{Q}$.
24:                 **return** "Start over."
25:      **return** "$\sigma < \mathsf{OPT}$".

---

**Theorem 8.** *Suppose that we are given a class of $n$ hypotheses, $\mathcal{H}$, a rate parameter $\eta \in (0, 1/4)$, two parameters $\sigma, \delta \in (0, 1)$, and we have access to $\hat{w}_j(H_i)$ with error at most $\epsilon'$ for every $i, j \in [n]$. Algorithm 5 queries $O\left(n \log(n/\delta)/\eta\right)$ many $\hat{w_i}(H_j)$ and runs in $O\left((n \log(n/\delta)/\eta) \cdot T_q\right)$ time. Also, it can be implemented in $O(1)$ rounds of adaptivity. This algorithm returns a hypothesis $H_\ell$, declares $\sigma < \mathsf{OPT}$, or returns "start over" for which the following guarantees hold with probability $1 - \delta$:*

- *If $\sigma \geq \mathsf{OPT}$, then the algorithm does not declare that $\sigma$ is less than $\mathsf{OPT}$.*

- *If the algorithm returns a hypotheses $H_\ell$, then $H_\ell$ is a $(\sigma + \epsilon', \, 3\sigma + 3\epsilon', \, m)$-seed where $m := \lfloor \eta \cdot n \rfloor$.*

- *If the algorithm returns "start over." with probability $1 - \delta$, it marks $(1 - 2\eta)$ fraction of the hypotheses in $\mathcal{Q}$.*

*Proof.* Our first claim is that if $\sigma$ is at least $\mathsf{OPT}$, then we do not declare otherwise. This is due to the fact that the discrepancy between $H_{i^*}$ and $P$ on any subset of the domain, including the Scheffé sets, will not exceed $\mathsf{OPT}$. Hence, $\hat{w}_j(H_{i^*})$ is at most $\mathsf{OPT} + \epsilon' \leq \sigma + \epsilon'$, and we will not mark $H_{i^*}$ in Line 7, Line 10, nor Line 15. That is, for at least one hypothesis, i.e., $H_{i^*}$, the "if" statement in Line 16 holds. And, we return a seed or "start over." (And, we will never reach Line 25.)

Next, we show that if the algorithm outputs $H_i$, then it is a $(\sigma + \epsilon', \ 3\sigma + 3\epsilon', \ m)$-seed with high probability. First, observe that $\hat{W}(H_i)$ must be at most $3\sigma + 3\epsilon'$ due to the "if" condition in Line 18, satisfying one of the two conditions we need for our desired seed in Definition 3.1.

Now, we show the next required condition for $H_i$ to be a good seed holds as well: $|\mathcal{M}_{\sigma+\epsilon', 3\sigma+3\epsilon'}(H_i)| \leq m$. This condition requires that the number of $H_j$'s in $\mathcal{H}$ for which $\hat{w}_j(H_i)$ is in $(\sigma + \epsilon', 3\sigma + 3\epsilon']$ is bounded by $m$. We have already established that $\hat{w}_j(H_i)$ for every $j \in [n]$ is bounded by $3\sigma + 3\epsilon'$ since $\hat{W}(H_i)$ is at most $3\sigma + 3\epsilon'$. Hence, we only need to show that there are at most $m$ hypotheses $H_j$ such that $w_j(H_i)$ is larger than $\sigma + \epsilon'$.

Now, if there are more than $m$ hypothesis $H_j$ in $\mathcal{H}$ such that $\hat{w}_j(H_i) > \sigma + \epsilon'$. Now, if we sample $t := \lceil 8\,n \log{(2n/\delta)} / m \rceil$ hypotheses, we should observe at least one $H_j$ for which $\hat{w}_j(H_i)$ is larger than $\sigma + \epsilon'$, with probability at least $1 - \delta/(2n)$. However, we know that such a hypothesis was never observed because we did not mark $H_i$ earlier. By the union bound, with a probability of at least $1 - \delta/2$, no such $H_i$ exists. Hence, with probability at least $1 - \delta/2$, $H_i$ is an $(\sigma + \epsilon', 3\sigma + 3\epsilon', m')$-seed as promised in the statement of the lemma.

Next, assume we output "start over." after finding an unmarked hypothesis $H_i$. In this case, we know that $H_i$ is not marked and $\hat{W}(H_i) > 2\sigma + 3\epsilon'$. Note that we have sampled $t$ edges of $H_i$, and we never see $w_j(H_i) > \sigma + \epsilon'$. Using a very similar argument as we have above: one can show there cannot be more than $m' = \lceil \eta \cdot |Q| \rceil$ for which $w_j(H_i) > \sigma + \epsilon'$ with probability $1 - \delta/2$ (for every $H_i$). That means, we will mark $|\mathcal{Q}| - m'$ many hypothesis in $\mathcal{Q}$. If $|\mathcal{Q}| > t \geq 1/\eta$, this implies that $(1 - 2\eta)$ fraction of the hypothesis in $\mathcal{Q}$ are removed. If $|\mathcal{Q}| \leq t$, then we know all the hypothesis in $\mathcal{Q}$ have $w_j(H_i) \leq \sigma + \epsilon'$ and all of them will be marked.

Furthermore, we show that we did not wrongfully mark a hypothesis that was $\sigma$-close to $P$. Observe that the condition in Line 22 holds in two cases:

**Case 1: $\hat{w}_i(H_j) > \sigma + \epsilon'$.** It is straightforward to show that $H_j$ is not $\sigma$-close to $P$ since we have:

$$\|H_j - P\|_{\mathrm{TV}} \geq w_i(H_j) \geq \hat{w}_i(H_j) - \epsilon' > \sigma\,.$$

**Case 2: $\hat{w}_i(H_j) \leq \sigma + \epsilon'$ and $\hat{w}_j(H_i) \leq \sigma + \epsilon'$.** Even though $\hat{w}_\ell(H_j)$ is small in this case, we can indirectly deduce that $H_j$ is not $\sigma$-close to $P$. By the definition of $\hat{w}_j(H_i)$ and $\hat{w}_i(H_j)$, we have:

$$\|H_i - H_j\|_{\mathrm{TV}} = |H_i\,(\mathcal{S}_{i,j}) - H_j\,(\mathcal{S}_{i,j})| \leq |H_i\,(\mathcal{S}_{i,j}) - P\,(\mathcal{S}_{i,j})| - |P\,(\mathcal{S}_{i,j}) - H_j\,(\mathcal{S}_{i,j})|$$
$$= w_i(H_j) + w_j(H_i) \leq \hat{w}_i(H_j) + \hat{w}_j(H_i) + 2\epsilon' \leq 2\sigma + 2\epsilon'\,.$$

On the other hand, by the triangle inequality, we have:

$$\|P - H_j\|_{\mathrm{TV}} \geq \|P - H_i\|_{\mathrm{TV}} - \|H_i - H_j\|_{\mathrm{TV}} \geq W(H_i) - \|H_j - H_i\|_{\mathrm{TV}}$$
$$\geq \hat{W}(H_i) - \epsilon' - \|H_i - H_j\|_{\mathrm{TV}} > 3\sigma + 3\epsilon' - \epsilon' - (2\sigma + 2\epsilon') = \sigma\,.$$

Therefore, in both of the cases above, we do not mark an $\sigma$-close distribution to $P$.

By the union bound over all the steps, our guarantees hold with probability $1 - \delta$. In addition, it is not hard to see that the running time of the algorithm is $O\,((n \cdot t + n) \cdot T_q) = O(n \cdot (\log(n/\delta)) \cdot T_q/\eta)$ time.

**Rounds of adaptivity:** It is straight forward to show that this algorithm can be implemented with a constant round of adaptivity. Let $\mathcal{Q}_0$ denote the initial state of unmarked hypotheses at the algorithm's outset. Since hypotheses are marked in sequence, in round $i$, the set of unmarked hypotheses is $\mathcal{Q}_i := \mathcal{Q}_0 \cap H_i, H_{i+1}, \ldots, H_n$. As a result, we can agree on the set of random $H_j$'s in $\mathcal{Q}$ and $\mathcal{H}$ and request $\hat{w}_j(H_i)$ within a single round. If the size of $|\mathcal{Q}_i|$ is at most $t$, then we include all $\hat{w}_j(H_i)$ for every $i \in n$ and $H_j \in \mathcal{Q}$. Next, upon discovering an unmarked $H_i$, we initiate another round of adaptivity to request all $\hat{w}_j(H_i)$ and $\hat{w}_j(H_i)$ for all $j \in [n]$. This enables the implementation of the rest of the algorithm.

$\square$

## B.2 Boosting a seed

In this section, we provide an algorithm that receives a $(\kappa\,\sigma + \epsilon', (\kappa + 2)\sigma + 5\,\epsilon', m)$-seed and aims to find a $(\kappa'\,\sigma + \epsilon', (\kappa' + 2)\sigma + 5\,\epsilon', m')$-seed with smaller $m'$. That means, the size of $\mathcal{M}$ for this new seed is smaller which brings us closer to have enough time to fully investigate the set $\mathcal{M}$ for a future seed. Our algorithm runs in $\tilde{O}(n \cdot T_q/\eta)$ time where $\eta \approx m'/m$ is our rate parameter. This running time is adaptive to our budget. If we are aiming for $\tilde{O}(n)$ algorithm, we find a seed such that $m'$ is only smaller than $m$ by a constant factor (constant $\eta$); Alternatively, if we have more time, we can decrease $m'$ with a faster rate (smaller $\eta$).

While our main goal is to find a better seed, our algorithm may not always be able to do so; However, it makes progress towards the end goal of the algorithm one way or the other. There are three possible outcomes for our algorithm:

1. The algorithm finds an accurate enough $\hat{H}$ as the final output of the algorithm.

2. The algorithm finds a better seed as we aimed for.

3. The algorithm marks $(1 - 2\eta)$ fraction of the unmarked hypotheses; And after that, we start over the procedure.

Although we may have regressed in last case above, it does not happen too often. Since we have only $n$ unmarked hypotheses to begin with, and we marked a large fraction of them every time, we do not start more than $O(\log_{1/2\eta}(n))$ times over the course of the algorithm. Our approach is presented in Algorithm 6, and we prove its performance in Theorem 9.

---

**Algorithm 6** aims to find a better seed.

---

1: **procedure** BOOST-SEED($\mathcal{H}$, $\mathcal{Q}$, $H_i$, $\sigma$, $\kappa$, $\kappa'$, $\epsilon'$, $\delta$, $\eta$, and query access to $\hat{w}_j(H_i)$'s)
2: $\quad m' \leftarrow \lfloor \eta\, m \rfloor$
3: $\quad t \leftarrow \lceil 8\,n \log(2n/\delta)/m' \rceil$
4: $\quad$ **for** $H_j \in \mathcal{M}$ **do** $\qquad\qquad\qquad\qquad$ ▷ We denote $\mathcal{M}_{(\kappa\,\sigma+\epsilon',\ (\kappa+2)\sigma+5\,\epsilon')}(H_i)$ by $\mathcal{M}$.
5: $\quad\quad\quad \tilde{W}(H_j) \leftarrow$ COMPUTE-$\tilde{W}(\mathcal{H}, \mathcal{Q}, H_i, t)$
6: $\quad\quad H_\ell \leftarrow \arg\min_{H_j \in \mathcal{M}} \tilde{W}(H_j)$
7: $\quad\quad$ **if** $\tilde{W}(H_\ell) > \kappa'\,\sigma + \epsilon'$ **then**
8: $\quad\quad\quad$ Return $H_i$ as the final answer.
9: $\quad\quad$ **else**
10: $\quad\quad\quad$ Compute $\hat{W}(H_\ell)$.
11: $\quad\quad\quad$ **if** $\hat{W}(H_\ell) \leq (\kappa' + 2)\,\sigma + 5\,\epsilon'$ **then**
12: $\quad\quad\quad\quad$ Return $H_\ell$ as a seed.
13: $\quad\quad\quad$ **else**
14: $\quad\quad\quad\quad$ **for** $H_j \in \mathcal{H}$ **do**
15: $\quad\quad\quad\quad\quad$ **if** $\hat{w}_\ell(H_j) > \sigma + \epsilon'$ or $\hat{w}_j(H_\ell) \leq \kappa'\,\sigma + \epsilon'$ **then**
16: $\quad\quad\quad\quad\quad\quad$ Mark $H_j$ in $\mathcal{Q}$.
17: $\quad\quad\quad\quad$ **return** "Start over."

---

**Theorem 9.** *Suppose Algorithm 6 receives these parameters as input: $\sigma \in \mathbb{R}_{\geq 0}$, $\kappa$, $\kappa' \geq 1$, $\eta \in (0, 1/4)$, $\epsilon'$, $\delta \in (0, 1)$, and two non-negative integers $n$, $m$. Also, assume the algorithm has access to a class of $n$ hypotheses $\mathcal{H}$, and a set of unmarked hypotheses in $\mathcal{Q} \subseteq \mathcal{H}$. It also receives a hypothesis $H_i \in \mathcal{H}$. The algorithm has one of the three possible outcomes listed below. Now, for any setting of such input parameters, if $H_i$ is a $(\kappa\,\sigma + \epsilon', (\kappa + 2)\sigma + 5\,\epsilon', m)$ seed, then the following guarantees hold for the outcome of Algorithm 6 with a probability of at least $1 - \delta$:*

1. *If the algorithm outputs a hypothesis $\hat{H}$ as the final answer, then we have for every $\epsilon \geq 6\,\epsilon'$:*

$$\|\hat{H} - P\|_{TV} \leq \max\left(\kappa + 2, \frac{\kappa + 2}{\kappa'} + 2\right) \cdot \max(\sigma, \mathsf{OPT}) + \epsilon.$$

2. *If the algorithm outputs a hypothesis $H_\ell$ as a new seed, then $H_\ell$ is a $(\kappa'\,\sigma + \epsilon', (\kappa' + 2)\sigma + 5\,\epsilon', m')$-seed where $m' := \lfloor \eta\, m \rfloor$.*

---

**Algorithm 7** computes $\tilde{W}(H_i)$.

---

1: **procedure** COMPUTE-$\tilde{W}$($\mathcal{H}, \mathcal{Q}, H_i, t$, and query access to $\hat{w}_j(H_i)$'s)
2:      $\tilde{W} \leftarrow 0$
3:      **repeat** $t$ **times**
4:          $H_j \leftarrow$ a uniformly random sample drawn from $\mathcal{H}$
5:          $\tilde{W} \leftarrow \max\left(\tilde{W}, \hat{w}_j(H_i)\right)$
6:          $H_j \leftarrow$ a uniformly random sample drawn from $\mathcal{Q}$
7:          $\tilde{W} \leftarrow \max\left(\tilde{W}, \hat{w}_j(H_i)\right)$
8:
9:      **if** $|\mathcal{Q}| \leq t$ **then**
10:          **for** all $H_j \in \mathcal{Q}$ **do**
11:             $\tilde{W} \leftarrow \max\left(\tilde{W}, \hat{w}_j(H_i)\right)$
12:      Return $\tilde{W}$.

---

    *3. If the algorithm requires us to start over, then we have marked at least $(1 - 2\eta)$ of the unmarked hypotheses.*

*The algorithm runs in $O(n \cdot (\log(n/\delta)) \cdot T_q/\eta)$ time and can be implemented in $O(1)$ rounds of adaptivity.*

*Proof.* Let $\mathcal{M}$ denote the set $\mathcal{M}_{\kappa\,\sigma+\epsilon',\,(\kappa+2)\sigma+5\,\epsilon'}(H_i)$. We consider the three possible outcomes of the algorithm, and prove the algorithm satisfied the desired guarantee in each of the three cases.

**Case 1: $\tilde{W}(H_\ell) > \kappa'\sigma + \epsilon'$.** In this case, the algorithm will return $H_i$ as the final answer. We show that the total variation distance between $H_i$ and $P$ is as desired. There are two possibilities depending on whether $H_{i^*}$ is in $\mathcal{M}$ or not.

**Case 1.1: $H_{i^*} \in \mathcal{M}$.** Using that $H_\ell$ has the minimum $\tilde{W}(H_\ell)$ among all the hypotheses in $\mathcal{M}$, we can bound OPT as follows:

$$\mathsf{OPT} \geq W(H_{i^*}) \geq \hat{W}(H_{i^*}) - \epsilon' \geq \hat{W}(H_\ell) - \epsilon' \geq \tilde{W}(H_\ell) - \epsilon' \geq \kappa'\sigma\,.$$

We prove that $H_i$ is not too far from $P$. Note that since $H_i$ was an $(\kappa\,\sigma + \epsilon',\,(\kappa+2)\,\sigma + 5\,\epsilon',\,m)$-seed, we are guaranteed that $\hat{W}(H_i)$ is bounded by $(\kappa+2)\,\sigma + 5\,\epsilon'$. Using the above bound for OPT and Fact C.3, we get:

$$\|H_i - P\|_{\mathrm{TV}} \leq W(H_i) + 2\,\mathsf{OPT} \leq \hat{W}(H_i) + \epsilon' + 2\,\mathsf{OPT} \leq (\kappa+2)\,\sigma + 6\,\epsilon' + 2\,\mathsf{OPT}$$

$$\leq \left(\frac{\kappa+2}{\kappa'} + 2\right) \cdot \mathsf{OPT} + 6\,\epsilon' \leq \left(\frac{\kappa+2}{\kappa'} + 2\right) \cdot \max(\sigma, \mathsf{OPT}) + \epsilon\,.$$

**Case 1.2: $H_{i^*} \notin \mathcal{M}$.** Since $H_i$ is a $(\kappa\,\sigma + \epsilon',(\kappa+2)\sigma + 5\,\epsilon',m)$-seed, $\hat{w}_{i^*}(H_i)$ is bounded by $\hat{W}(H_i) \leq (\kappa+2)\sigma + 5\,\epsilon'$. Now that $H_{i^*}$ is not in $\mathcal{M}$, it must be the case that $\hat{w}_{i^*}(H_i)$ is bounded by $\kappa\,\sigma + \epsilon'$. In this case, we have the following bound for the total variation distance between $H_i$ and $P$ via Fact C.3:

$$\|H_i - P\|_{\mathrm{TV}} \leq w_{i^*}(H_i) + 2\,\mathsf{OPT} \leq \hat{w}_{i^*}(H_i) + \epsilon' + 2\,\mathsf{OPT} \leq \kappa\,\sigma + 2\,\epsilon' + 2\,\mathsf{OPT}$$

$$\leq (\kappa+2)\max(\sigma, \mathsf{OPT}) + \epsilon\,.$$

**Case 2: $\tilde{W}(H_\ell) \leq \kappa'\sigma + \epsilon'$ and $\hat{W}(H_\ell) \leq (\kappa'+2)\,\sigma + 5\,\epsilon'$.** In this case, we output $H_\ell$ as a new seed. Given the assumptions of this case, we show that $H_\ell$ is a $(\kappa'\sigma + \epsilon',(\kappa'+2)\sigma + 5\,\epsilon',m')$-seed. First, $\hat{W}(H_\ell)$ is at most $(\kappa'+2)\,\sigma + 5\,\epsilon'$ implying the first desired property for being an $(\kappa'\sigma + \epsilon',(\kappa'+2)\sigma + 5\,\epsilon',m')$-seed holds for $H_\ell$. Next, we show the second desired property for $H_\ell$: the size of $\mathcal{M}_{\kappa'\,\sigma+\epsilon',\,(\kappa'+2)\sigma+5\,\epsilon'}(H_\ell)$ is bounded by $m'$. At a high level, we expect the size of

this set to be small; Otherwise, we would have observed one of these large $\hat{w}_\ell(H_j)$ when we have sampled $t$ hypotheses to compute $\tilde{W}(H_\ell)$.

Suppose $H_\ell$ has more than $m'$ edges with $\hat{w}_j(H_\ell) > \kappa'\sigma + \epsilon'$. Now, if we sample $t :=$ $\lceil 8\, n \log (2n/\delta) / m' \rceil$ hypotheses, we should observe at least one $H_j$ for which $\hat{w}_j(H_\ell)$ is larger than $\kappa'\sigma + \epsilon'$, with probability at least $1 - \delta/(2n)$. However, we know that such a hypothesis was never observed because $\tilde{W}(H_\ell)$, which is the maximum of $\hat{w}_j(H_\ell)$'s, is bounded from above by $\kappa'\sigma + \epsilon'$. By the union bound, with a probability of at least $1 - \delta/2$, no such $H_\ell$ exists. Hence, with probability at least $1 - \delta/2$, $H_\ell$ is an $(\kappa'\sigma + \epsilon', (\kappa'+2)\sigma + 5\epsilon', m')$-seed as promised in the statement of the lemma.

**Case 3: $\tilde{W}(H_\ell) \leq \kappa'\sigma + \epsilon'$ and $\hat{W}(H_\ell) > (\kappa'+2)\sigma + 5\epsilon'$.** In this case the algorithm marks any unmarked hypothesis $H_j$ for which $\hat{w}_\ell(H_j) > \sigma + \epsilon'$ or $\hat{w}_j(H_\ell) \leq \kappa'\sigma + \epsilon'$, and we start over. First, we show that we did not wrongfully mark a hypothesis that was $\sigma$-close to $P$. Observe that the condition in Line 15 holds in two cases:

**Case 3.1: $\hat{w}_\ell(H_j) > \sigma + \epsilon'$.** It is straightforward to show that $H_j$ is not $\sigma$-close to $P$ since we have:
$$\|H_j - P\|_{\mathrm{TV}} \geq w_\ell(H_j) \geq \hat{w}_\ell(H_j) - \epsilon' > \sigma\,.$$

**Case 3.2: $\hat{w}_\ell(H_j) \leq \sigma + \epsilon'$ and $\hat{w}_j(H_\ell) \leq \kappa'\sigma + \epsilon'$.** Even though $\hat{w}_\ell(H_j)$ is small in this case, we can indirectly deduce that $H_j$ is not $\sigma$-close to $P$. By the definition of $\hat{w}_j(H_\ell)$ and $\hat{w}_\ell(H_j)$, we have:
$$\|H_\ell - H_j\|_{\mathrm{TV}} = |H_\ell\,(\mathcal{S}_{\ell,j}) - H_j\,(\mathcal{S}_{\ell,j})| \leq |H_\ell\,(\mathcal{S}_{\ell,j}) - P\,(\mathcal{S}_{\ell,j})| - |P\,(\mathcal{S}_{\ell,j}) - H_j\,(\mathcal{S}_{\ell,j})|$$
$$= w_\ell(H_j) + w_j(H_\ell) \leq \hat{w}_\ell(H_j) + \hat{w}_j(H_\ell) + 2\,\epsilon' \leq \kappa'\sigma + \sigma + 4\,\epsilon'\,.$$

On the other hand, by the triangle inequality, we have:
$$\|P - H_j\|_{\mathrm{TV}} \geq \|P - H_\ell\|_{\mathrm{TV}} - \|H_\ell - H_j\|_{\mathrm{TV}} \geq W(H_\ell) - \|H_j - H_\ell\|_{\mathrm{TV}}$$
$$\geq \hat{W}(H_\ell) - \epsilon' - \|H_\ell - H_j\|_{\mathrm{TV}} > (\kappa'+2)\sigma + 4\,\epsilon' - ((\kappa'+1)\sigma + 4\,\epsilon') = \sigma\,.$$

Therefore, in both of the cases above, we do not mark an $\sigma$-close distribution to $P$.

Second, we claim that we mark at least $(1 - 2\,\eta)$ fraction of the unmarked hypotheses in $\mathcal{Q}$. Recall that when we compute $\tilde{W}(H_\ell)$ we also sample $t$ hypotheses from the set of unmarked hypotheses, and we did not observe any edge $\hat{w}_j(H_\ell)$ that is larger than $\kappa'\sigma + \epsilon'$. With a very similar argument we had above, with probability $1 - \delta/(2n)$, we do not have more than $m'' := \lceil \eta \cdot |\mathcal{Q}| \cdot m/n \rceil \leq \lceil \eta \cdot |Q| \rceil$ hypotheses in $\mathcal{Q}$ such that $\hat{w}_j(H_\ell) > \kappa'\sigma + \epsilon'$. Otherwise, we would have seen one of these edges, and $\tilde{W}(H_\ell)$ would have been larger. By the union bound, this fact holds for every $H_\ell$. Hence, the "if" condition in Line 15 holds for over $|\mathcal{Q}| - \lceil \eta \cdot |\mathcal{Q}| \rceil$ of the unmarked hypotheses in $\mathcal{Q}$, and the algorithm mark them. If $|\mathcal{Q}| \geq 1/\eta$, it is easy to show that $|\mathcal{Q}| - \lceil \eta \cdot |\mathcal{Q}| \rceil$ is at least $(1 - 2\,\eta) \cdot |\mathcal{Q}|$. Thus, we mark at least $(1 - 2\,\eta)$ fraction of the hypotheses in $\mathcal{Q}$ as we have claimed. Now, if $|Q| \leq 1/\eta$, it is easy to show that $|Q| < t$. Hence, the algorithm involves all the $H_j$'s in $\mathcal{Q}$ to compute $\hat{W}(H_\ell)$. Therefore, every single $\hat{w}_j(H_\ell)$ is at most $\kappa'\sigma + \epsilon'$ and we mark all the hypotheses in $\mathcal{Q}$.

By the union bound over all the steps, our guarantees hold with probability $1 - \delta$. In addition, it is not hard to see that the running time of the algorithm is $O\,((m \cdot t + n) \cdot T_q) = O(n \cdot (\log(n/\delta)) \cdot T_q/\eta)$ time.

**Rounds of adaptivity:** Our algorithm can be implemented in constant rounds of adaptivity. To compute $\tilde{W}$'s, we can preselect the random $H_j$'s. If $\mathcal{Q}$ contains fewer than $t + 1$ hypotheses, we include every $\hat{w}_j(H_i)$ for each $H_j \in \mathcal{Q}$ and $i \in [n]$. Upon discovering $H_\ell$, another round of adaptivity allows us to query every $\hat{w}_j(H_\ell)$ and $\hat{w}_\ell(H_j)$ to calculate $\hat{W}(H_\ell)$ and mark the hypotheses in $\mathcal{Q}$.

$\square$

## C  Preliminary facts and lemmas

**Fact C.1.** *For all distribution $P$ and for all probability event $\mathcal{E}$ defined under $P$, we have:*

$$\|P - P_{|\mathcal{E}}\|_{TV} = 1 - P(\mathcal{E}),$$

*where $P_{|\mathcal{E}}$ denotes the probability distribution $P$ when conditioned on the event $\mathcal{E}$.*

**Fact C.2.** *For every pair of distributions $H_i$ and $H_j$, we have:*

$$\|H_i - H_j\|_{TV} = |H_i(\mathcal{S}_{ij}) - H_j(\mathcal{S}_{ij})|.$$

The following fact is adapted from [DL01, MS08, ABS23].

**Fact C.3.** *Suppose $H_{i^*}$ is the closest hypothesis to $P$ in $\mathcal{H}$(i.e., $\|H_{i^*} - P\|_{TV}$). For every pair of hypotheses $H_i$ and $H_j$, the following holds:*

    *1.* $\|H_i - P\|_{TV} \leq w_j(H_i) + 2\|H_j - P\|_{TV}$,

    *2.* $\|H_i - P\|_{TV} \leq w_{i^*}(H_i) + 2\,\mathsf{OPT}$,

    *3.* $\|H_i - P\|_{TV} \leq W(H_i) + 2\,\mathsf{OPT}$.

*Proof.* The proof is based on triangle inequality, and the definitions of $w_j(H_i)$'s and $\mathsf{OPT}$. For every $H_j$ in $\mathcal{H}$, we have:

$$
\begin{aligned}
\|H_i - P\|_{TV} &\leq \|H_i - H_j\|_{TV} + \|H_j - P\|_{TV} = |H_i(\mathcal{S}_{i,j}) - H_j(\mathcal{S}_{i,j})| + \|H_j - P\|_{TV} \\
&\leq |H_i(\mathcal{S}_{i,j}) - P(\mathcal{S}_{i,j})| + |P(\mathcal{S}_{i,j}) - H_j(\mathcal{S}_{i,j})| + \|H_j - P\|_{TV} \\
&= w_j(H_i) + w_i(H_j) + \|H_j - P\|_{TV} \\
&\leq w_j(H_i) + 2\|H_j - P\|_{TV}.
\end{aligned}
$$

Hence, Item 1 is proved. Now, if we set $H_j$ to be $H_{i^*}$, then $\|H_{i^*} - P\|_{TV}$ is equal to $\mathsf{OPT}$ implying Item 2. Item 3 is concluded from Item 2 and the fact that $w_{i^*}(H_i)$ is upper bounded by $W(H_i)$. $\square$

**Fact C.4.** *Suppose we are given six parameters $a, b, a', b' \in \mathbb{R}_{\geq 0}$, and $m, m' \in \mathbb{Z}_{\geq 0}$. If $a \leq a'$, $b \leq b'$, and $m \leq m'$, every $(a, b, m)$-seed is also an $(a', b', m')$-seed.*

**Fact C.5.** *Suppose we have a set of $n$ hypotheses $\mathcal{H}$, and a predicate, $R(H) : \mathcal{H} \to \{0, 1\}$, that maps the hypotheses in $\mathcal{H}$ to zero or one. Assume $\mathcal{H}$ contains more than $m$ hypotheses with $R(H) = 1$ for an arbitrary integer parameter $0 \leq m < n$. If we draw $s \geq 8\,n \log(\delta^{-1})/m$ hypotheses from $\mathcal{H}$ uniformly at random, we will observe at least one hypothesis with $R(H) = 1$ with probability at least $1 - \delta$.*

*Proof.* Let $p$ denote the fraction of such hypothesis in $\mathcal{H}$ with $R(H) = 1$. Observe that $p > 0$ since there are at least $m + 1$ such hypothesis in $\mathcal{H}$. Then, using the Chernoff bound, we have:

$$
\begin{aligned}
\mathbf{Pr}[r = 0] &\leq \mathbf{Pr}\left[r < \frac{s\,p}{2}\right] = \mathbf{Pr}\left[\frac{r}{s} < \left(1 - \frac{1}{2}\right) \cdot p\right] \\
&\leq \exp\left(-\frac{s\,p}{8}\right) < \exp\left(-\frac{s\,m}{8\,n}\right) \leq \delta.
\end{aligned}
$$

$\square$

## D  Data structure for marked and unmarked hypotheses

Here, we describe a simple data structure that allows us to efficiently keep track of marked and unmarked hypotheses. Our data structure consists of two integers, `n` and `t`, and two arrays of size `n`: `index[]` and `list[]`. Here, `n` represents the number of hypotheses that the data structure supports. The `list` array is an arbitrary ordering of integers from `1` to `n`. Throughout the operations of this data structure, we preserve the following guarantee: the value of `index[i]` indicates where to find element `i` in the `list`. In other words, `list[index[i]]` is always `i`. The integer `t` serves as a threshold quantity. The first `t` numbers in the list correspond to unmarked hypotheses, while the remaining numbers represent marked hypotheses. Initially, `list` is a list of integers from `1` to `n` in

ascending order. And, for every hypothesis $H_i$, index[i] is set to i. This data structure supports the following operations:

- Initialization ds(size): To initialize the data structure, we set n and t to size. Next, we set list to be a list of integers from 1 to n in ascending order. To ensure consistency with the list, each index[i] is set to i.

- mark(i): To mark hypothesis $H_i$, we swap the element at index i with the element at index t in the list array, and then decrement t by 1.

- is_marked(i): Given our definition, if index[i] is less than or equal to t, then $H_i$ is unmarked; otherwise, it is marked.

- is_all_marked(): If t is equal to zero, then it means all the hypotheses are marked.

- random_unmarked(): To select a random unmarked hypothesis, we generate a random integer r between 1 and t, and output the hypothesis at list[r].

- random_marked(): To select a random marked hypothesis, we follow the same process as for an unmarked hypothesis, except that r is generated between t+1 and n.

It is easy to see that the initialization takes $O(n)$ time, and the rest of the operations only take $O(1)$ time.

