# OpenReview forum: "Optimal Hypothesis Selection in (Almost) Linear Time"
_NeurIPS.cc/2024/Conference — NeurIPS 2024 poster_

### Official Review · Reviewer_reSf · 2024-07-03

**Soundness:** 3
**Presentation:** 3
**Contribution:** 3
**Rating:** 8
**Confidence:** 3

**Summary:**

This paper studies the hypothesis selection problem: Given $n$ distributions $H_1, H_2, \ldots, H_n$ and samples from an unknown distribution $P$, the goal is to output $\hat H$ such that $\mathrm{TV}(P, \hat H) \le \alpha \cdot \mathsf{OPT} + \epsilon$, where $\mathsf{OPT} = \min_{i \in [n]}\mathrm{TV}(P, H_i)$ is the TV-distance of the best hypothesis.

It is known that achieving a guarantee with $\alpha < 3$ is impossible, if we want the sample complexity to be bounded by a function of $n$ and $\epsilon$ (and not the domain size). Prior work gave various algorithms that achieve $\alpha = O(1)$ (including some with $\alpha = 3$), but none of them achieves $\alpha = 3$ in near-linear time without making additional assumptions.

This work gives two improved algorithms. The first algorithm (Algorithm 1) achieves $\alpha = 3$ in $\tilde O(n\cdot s / \epsilon)$ time, where $s = \Theta((\log n) / \epsilon^2)$ is the sample complexity. The second algorithm (Algorithm 4) achieves $\alpha = 4$ in $\tilde O(n\cdot s)$ time.

Here is an overview of Algorithm 1, outlined by the authors in Section 3:
- Let $S_{i j}$ denote the subset of the domain that witnesses $\mathrm{TV}(H_i, H_j)$. A simple argument shows that finding $i \in [n]$ that minimizes the quantity $W(H_i) \coloneqq \max_{j \in [n]}|H(S_{i j}) - P(S_{i j})|$ gives the $\alpha = 3$ guarantee. However, the straightforward implementation takes $\Omega(n^2)$ time to compute $W(H_i)$ for all $i \in [n]$.
- The actual algorithm maintains estimates on $W(H_i)$s (that always underestimate). These are grouped into $\approx 1/\epsilon$ "buckets", where bucket $l$ contains indices with estimated $W(H_i) \approx l\cdot\epsilon$.
- In each iteration, we find the lowest non-empty bucket $l$, and tries to update the estimates. Concretely, we loop over all $j \in [n]$ and only sample a few indices $i$ from bucket $l$. For each $(i, j)$ pair, we try to update the estimate of $W(H_i)$ using $w_j(H_i)$. This might bump some $i$ into larger buckets.
- If there is a single $j \in [n]$ that bumps a substantial fraction of the sampled $i$s, we call such $j$ a "prompting hypothesis", and we use such $j$ to update all estimates in the current bucket. Note that if we keep finding prompting hypotheses, we will empty a bucket in $O(\log n)$ repetitions, and the whole process must end in $O((\log n)/\epsilon)$ iterations. Moreover, each iteration is almost linear-time.
- The harder part is to argue that, whenever we cannot find a prompting hypothesis, we are done. At a high level, this boils down to showing that the optimal hypothesis $i^*$ must be prompting, unless the majority of hypothesis in the lowest bucket are already good enough.

**Strengths:**

- This work makes progress on improving the runtime for a fundamental problem in statistics / machine learning.
- The new algorithms are based on clever ideas and several new concepts, which might find applications in future work.
- The writing is very clear and I found the main paper relatively easy to follow. I especially liked that the authors spent most of the main paper on the proof overview, which is enjoyable to read and should be able to convince the readers why the proposed methods work.

Overall, this paper contains fairly significant results that are very well presented and should be of interest to most learning theorists. I vote for acceptance.

**Weaknesses:**

The main weakness is that this work is unlikely to have said the last words on the problem---it remains open whether one can get $\alpha = 3$ in $\tilde O(n/\epsilon^2)$ time.

Minor writing suggestions:
- Equation (2): Missing extra space in $S_{ij}$ on the left-hand side? (The notational convention in the rest of the paper seems to be $S_{i j}$, though personally writing $S_{i,j}$ might be clearer.)
- Line 90: "a $O(\cdots)$ factor" --> "an $O(\cdots)$ factor"
- Line 270: "dependency" not capitalized.

**Questions:**

Is there a reason why the "right" runtime should be $n \cdot s = \tilde O(n/\epsilon^2)$? For example, given that the algorithm needs $\Omega(n + s) = \tilde\Omega(n + 1/\epsilon^2)$ time to read the inputs---can we hope to get an $\tilde O(n + 1/\epsilon^2)$ runtime?

**Limitations:**

Limitations are adequately addressed in Section 1.2.

---

> ### Author Rebuttal · Authors · 2024-08-07
>
> Thank you for your feedback.
>
> In response to your question, there is no explicit lower bound for the time complexity of this problem. As you mentioned, it is possible  that an algorithm could exist with a time complexity of just $\Theta(n + s)$. However, for algorithms that operate by querying the probabilities of various sets according to an unknown distribution $P$—such as querying semi-distances or similar concepts, as many previous algorithms have done—we speculate that $\Theta(n)$ queries to the semi-distances are both necessary and sufficient.
> This would result in an overall time complexity of $\Omega(n \cdot s)$. We will clarify this in the future versions of our paper.
>
> Evidence suggesting that $O(n)$ queries might be sufficient comes from the upper bound of [MS08]. They demonstrated that there exist $O(n)$ sets to query, whose responses are sufficient to solve the problem. In their work, these sets and the order in which to query them are determined through an exponential time preprocessing step over the class $\mathcal{H}$. It would be valuable to explore whether this problem can be solved with $\Theta(n)$ queries without relying on the preprocessing step.

---

> > ### Comment · Reviewer_reSf · 2024-08-07
> >
> > Thank you for your reply! I have no further questions, and my overall evaluation of the paper remains positive.

---

### Official Review · Reviewer_URPo · 2024-07-11

**Soundness:** 3
**Presentation:** 3
**Contribution:** 3
**Rating:** 7
**Confidence:** 3

**Summary:**

This paper studies the hypothesis selection problem, where given a set of candidate distributions $\mathcal{H}=\{H_1,\dots,H_n\}$ and samples from a distribution $P$, the learner wants to approximately select a distribution $H_i \in \mathcal{H}$ such that $||H_i-P||_{TV} \le \alpha opt+\epsilon$ for some constant $\alpha$, where $opt$ is the TV distance of the closest distribution in $\mathcal{H}$ to $P$. Information theoretically, $\alpha=3$ is the optimal constant one could achieve with optimal sample complexity $s=\log n/\epsilon^2$. Previous efficient algorithms either have a running time that has a bad dependence on $n$ or run in nearly linear time but achieve a suboptimal approximate factor $\alpha \ge 5$. In this paper, the authors design an algorithm that achieves the optimal approximate factor $\alpha=3$, with running time $\tilde{O}(ns/\epsilon)$. This is the first algorithm that achieves the optimal approximate factor with a running time nearly linearly dependent on $n$. This paper also presents another algorithm that achieves an approximate factor $\alpha = 4$ but has a better running time $\tilde{O}(ns)$

**Strengths:**

This paper makes progress on a fundamental problem, hypothesis selection. The two algorithms designed by the authors are highly non-trivial and involve many novel techniques. The paper is in general written well and gives a good intuition of why the algorithms work. Due to the time limit, I was not able to check the proofs in the appendix but given the discussions in the main body, I believe the results given by the paper are correct.

**Weaknesses:**

I want to give some minor suggestions.

1. It would be nice to give some more applications or motivations at the beginning of the introduction. In the current version of the paper, the problem sounds mathematically interesting but lacks motivation.

2. Since there is a long line of works that design nearly linear time algorithms for the problem. It would be helpful to give a brief review of the techniques used in prior works in the introduction.

3. In line 327, I believe there is a typo in the equation $||H_{i^*}||_{TV}=...$

**Questions:**

I only have one question about the model. It seems that the algorithms in the paper crucially rely on estimating the semi-distance defined based on the Scheffé set. But I believe in many applications it would be very hard to check whether an example falls into the Scheffé set of two hypothesises. For example, many learning algorithms might first improperly learn a list of candidate hypothesises and use hypothesis selection to select a good one. Computing pdf of these candidate hypothesises may not be realistic in general. Would it be possible to relax the dependence on the knowledge of the pdf of the candidate hypothesises in the model?

---

> ### Author Rebuttal · Authors · 2024-08-07
>
> Thank you for your feedback. We will include the motivations and applications of hypothesis selection, as well as an overview of previous algorithms, in our paper. In this rebuttal, some applications of hypothesis selection are discussed in our response to Reviewer fMmw, for your reference.
>
> Regarding your question about relaxing the model, the short answer is yes; our algorithms remain effective with some minimal adjustments as long as approximations of Scheffe sets are provided.
>
> Currently, in our paper, we compare the PDF of the known distributions, $H_i(x) > H_j(x)$, to determine whether $x$ belongs to $S_{ij}$, the Scheffe set of $H_i$ and $H_j$. However, this assumption can be relaxed if we can identify an alternative set $S'\_{ij}$ that captures most of the discrepancy between $H_i$ and $H_j$. In particular, it is sufficient to have:
> $$\|H\_i - H\_j \|\_{TV} \leq (1+ \Theta( \epsilon)) \cdot |H_i(S'\_{ij}) - H_j(S'\_{ij}) | \,.$$
> Even with these imperfect sets, $w_{i^*}(H_i)$ is a good proxy for the quality of hypothesis $H_i$ (up to additive error $\epsilon$). See the equations after Line 180 in the paper.

---

> > ### Comment · Reviewer_URPo · 2024-08-09
> >
> > Thanks for the response.

---

### Official Review · Reviewer_Zrpk · 2024-07-11

**Soundness:** 3
**Presentation:** 1
**Contribution:** 3
**Rating:** 5
**Confidence:** 3

**Summary:**

This paper introduces proper approximation algorithms for optimal hypothesis selection in the context of distribution learning.
The first algorithm achieves an optimal approximation factor of $\alpha=3$ in time approximately linear in the number of hypotheses. The second achieves the slightly looser $\alpha = 4$ approximation factor in exchange for significant reduced dependence
on $\epsilon$ and $\delta$, the standard learning parameters governing the error and confidence of the learning algorithms. The paper focuses on explanation of the algorithmic techniques involved.

**Strengths:**

The paper introduces a novel learning algorithm for optimal approximation (Algorithm 1) which achieves a significantly reduced dependence (quadratic to near-linear) on the number of hypotheses in the hypothesis class in the computational complexity. This is notable given that the problem has been studied for a while, and practical, given that distribution learning over a discrete space requires a number of samples at least proportional to the cardinality of the domain on which the hypotheses are supported.

**Weaknesses:**

I found the writing to be unpolished, and not publication-ready. Much of the body attempts to furnish intuition to the algorithmic techniques introduced, but some portions seem underdeveloped, and occasionally read like direct translations from math to spoken language (e.g. line 358). For example, an extremely important concept in this paper (introduced in important prior works) is that of ``semi-distances'' $w_i(H_j)$. Seemingly the best intuition the reader is provided with regarding this concept comes at line 176: "One suggestion for readers to internalize the semi-distances is to view them as a distance between $H_i$ and $P$ that is measured from the perspective of $H_j$". I think this deserves more illuminating wording, and personally got a much better understanding by staring longer at the definition. Overall, I think the lack of attention to writing is somewhat a shame, as there seems to be a lot of nice algorithmic thought here which deserves a better exposition in my opinion.

On a more technical level, the dependence in the computational complexity of Algorithm 1 on the learning parameters $\epsilon$ and $\delta$ is heavy -- these enter as $1/\delta^3\epsilon^3$. One would really hope for something like $\log(1/\delta)polylog(1/\epsilon)/\epsilon^2$.  It seems there will be many regimes in which the guarantees of Algorithm 1 -- despite the linear dependence on the number of hypotheses -- will be looser than the quadratic algorithm of MS08.

**Questions:**

-Table 1: Is the dependence on $1/\delta$ in $s$ just the standard $\log(1/\delta)$?

-line 180: The order of logical quanitifiers can be reversed to get a slightly stronger and more illustrative statement here, correct?

-line 240: It seems that to do this random sampling to decide if you have a prompting hypothesis, you need to check a number of hypotheses which is proportional to $1/\delta^2$. I guess there is a naiver version of this algorithm which just checks all of the hypotheses at this step, incurring some quadratic dependence in $n$. Am I wrong in feeling like any sort of search for something like a promping hypotheses will always incur some sort of undesirable multiplicative interaction with $n$?

**Limitations:**

Yes

---

> ### Author Rebuttal · Authors · 2024-08-07
>
> **Presentation:**
> Thank you for your feedback regarding the presentation of our paper. Our overview was intended to provide a high-level description of our algorithm to avoid obscuring the main technical ideas with detailed specifics. We will certainly focus on enhancing the clarity and quality of our write-up in future versions.
>
> **Dependency on $\epsilon$ and $\delta$:** Please see our global rebuttal.
>
> Below is our answer to your questions:
>
> - In Table 1, for [ABS23] the dependence on $\delta$ is $1/\delta$, while for the rest it is $O(\log(1/\delta))$ (or unspecified by the authors).
>
> - Correct. There exists an $i^*$ such that for every $H_i$, $w_{i^*}(H_i)$ determines the quality of $H_i$.
>
> - Correct. To ensure that a random hypothesis in a bucket is not too far, with a probability of $1-\delta$ in a single round, we have to sample $O(\log(n)/\delta)$ hypotheses and check if $H_{i^}$ is prompting them. Or, we can try all hypotheses in the bucket. Hence, relying on this structural property of $H_{i^*}$ makes a polynomial dependency on $\delta$ inevitable. We speculate that improving the dependency on $\delta$ for this algorithm would require new algorithmic ideas. The main difficulty here is that there is no (known) general technique for boosting the confidence parameter while keeping $\alpha$ the same. In many settings, the success probability of a learning algorithm can be amplified from a constant, say 2/3, to at least $1-\delta$ at a cost of at most $\log(1/\delta)$ in running time and sample complexity. However, in hypothesis selection, choosing the best output from several runs of a given algorithm requires executing a second hypothesis selection algorithm, which introduces another factor of $\alpha$ in the approximation—leading to a total factor of at least $9$. As a result, these kinds of two-phase algorithms are not sufficient in the low $\alpha$ regime. Some previous results, such as [ABS23], also suffer from a polynomial dependency on $\delta$. Our second algorithm circumvents this polynomial dependence at the cost of a slightly worse accuracy parameter $(\alpha = 4)$.

---

> > ### Comment · Reviewer_Zrpk · 2024-08-09
> >
> > Thanks for the reply. I do have a concern that the algorithmic techniques for the $\alpha=3$ case may not be very informative for future algorithmic development re: the discussion on prompting hypotheses above. However, in retrospect, I think this is probably a bit unfair, as I'm very far removed from the study of this particular problem. Thus, I revise my score to 5.
> >
> > I do strongly encourage the authors do improve the presentation in the next version. Good work deserves good presentation.

---

> > > ### Author Response · Authors · 2024-08-11
> > >
> > > Thank you for your comment and for raising your score.
> > >
> > > We understand your concern regarding the direct usability of our algorithms in future work. While we cannot guarantee their applicability, our hope is that the novel perspective we introduced, along with the new algorithmic ideas, will lead to improvements in the time complexity of this problem. We will invest more effort in distilling the new algorithmic ideas and structural results into a form that is widely useful.

---

### Official Review · Reviewer_fMmw · 2024-07-12

**Soundness:** 4
**Presentation:** 3
**Contribution:** 3
**Rating:** 6
**Confidence:** 4

**Summary:**

This paper looks at proper distribution learning: given samples from
some distribution p, and a set of n candidate distributions H_i,
output one H_i that is close in TV; in particular, alpha * OPT + eps.
Surprisingly, this is possible with a sample complexity independent of
the domain size (as would be needed to actually estimate the TV).
There's been a long line of work aiming to improve the approximation
factor alpha.  alpha = 2 is possible for *improper* learners, but
proper learners can only hope for alpha = 3.  Getting alpha=3 in n^2 s
time is known; this paper gets that down to O~(ns) time (although an
extra eps factor in time, which they can avoid for alpha=4, and worse
delta dependence).

**Strengths:**

It gives much better running time for a natural problem.

The approach is fairly clean.

**Weaknesses:**

The algorithm overview is a bit vague, and could be written more
clearly.

The failure probability dependence isn't good.

The alpha=3 approach is a fairly simple extension of prior work, and
alpha=4 isn't so exciting.

I'm not sure that the constant here matters for the applications of
hypothesis selection.  Like, in the application where we choose a
cover for the class, presumably we can just make a finer cover?

**Questions:**

Are there applications of hypothesis selection where the constant alpha matters?

**Limitations:**

Fine.

---

> ### Author Rebuttal · Authors · 2024-08-07
>
> **Presentation**
>
> Thank you for your feedback regarding the presentation of our paper. Our overview was intended to provide a high-level description of our algorithm to avoid obscuring the main technical ideas with detailed specifics. We will certainly focus on enhancing the clarity and quality of our write-up in future versions.
>
> **Novelty of our techniques**
>
> The essence of our first algorithm indeed stems from the minimum distance estimate [DL01]. However, the primary challenge we faced was implementing this approach in linear time. Specifically, it is not feasible to compute the maximum semi-distances, $W(H_i)$, for all hypotheses in linear time. To address this, we developed an efficient and advanced method to estimate these values by taking the maximum over a small subset of semi-distances, while ensuring the algorithm's correctness. This approach allows us to achieve an algorithm that runs in (almost) linear time in $n$.
>
> Our second algorithm is completely novel to the best of our knowledge. It achieves the desired time complexity (up to logarithmic factors) with $\alpha = 4$, an accuracy parameter that surpasses all previously known results. Our work introduces a novel algorithmic approach to this problem, marking a significant departure from existing techniques. We hope these new ideas will inspire future research.
>
> **Dependency on $\epsilon$ and $\delta$**
>
> Please see our global rebuttal.
>
>
> **Applications of hypothesis selection**
>
> Density estimation is a fundamental problem in statistics, with hypothesis selection being an important special case. It involves choosing the best distribution from a set of known models that represent potential underlying data distributions. For example, this set might include Poisson and gamma distributions with various parameters to model the number of arrivals per time unit. This technique is applicable in areas such as denoising, anomaly detection, selecting interpretable data distributions, strategy selection, and more.
>
> That said, we view our results as a fundamental theoretical tool. Hypothesis selection has been instrumental in learning structured distributions (e.g., learning mixture of Gaussians [DK14, SOAJ14]). For additional references, see Section 1.3. Another significant aspect of hypothesis selection is its agnostic nature, allowing for learning even when the unknown distribution is not within the considered class. Hence, hypothesis selection is applicable even when data is noisy.
>
>
> **Addressing the importance of improving $\alpha$ by a constant factor**
>
> In most learning algorithms, the error guarantee decreases polynomially as the number of samples increases, so constant factors may not be as crucial. However, this is not the case in hypothesis selection. The output hypothesis is guaranteed to be $(\alpha \cdot OPT + \epsilon)$-close to $P$ in total variation distance. While increasing the number of samples can reduce $\epsilon$ to negligible levels, it does not improve the term $\alpha \cdot OPT$. $\alpha$ is an inherent property of the algorithm and directly impacts the best achievable error guarantee. Therefore, even a constant improvement in $\alpha$ is significant.
>
> Some may argue that focusing on improving $OPT$ is more beneficial than refining $\alpha$. For instance, in the cover method (as you mentioned), using a finer $\gamma$-net can ensure that $OPT < \gamma$. However, this approach can drastically increase the algorithm's running time, as the size of the net can grow super-polynomially with $\gamma$. For example, in the case of mixtures of $k$ Gaussians, the dependence of the net size on $\gamma$ is roughly $O(\gamma^{-3\cdot k})$ (see [SOAJ14]). Thus, reducing $\gamma$ by a factor of three could increase the size of $\mathcal{H}$ by an exponential factor in $k$, and consequently, the running time, leading to an inefficient algorithm.

---

> > ### Comment · Reviewer_fMmw · 2024-08-09
> >
> > Thank you for your response.

---

### Author Rebuttal · Authors · 2024-08-07

Thank you for taking the time to review our paper and for your feedback.

**Presentation:**
Thank you for your comments regarding the presentation of our paper. We will incorporate all your editorial suggestions regarding the presentation of the paper. We will certainly focus on enhancing the clarity and quality of our write-up in future versions.

**Dependency on $\epsilon$ and $\delta$:**
We acknowledge that, ideally, an algorithm should achieve a running time of $O(n \log(1/\delta)/\epsilon^2)$. Our first algorithm, with $\alpha = 3$, does not meet this ideal due to suboptimal dependencies on $\delta$ and $\epsilon$. However, it marks a significant step forward as it is the first in two decades to achieve a time complexity linear in $n$ for any $\alpha < 5$. To address the shortcomings, our second algorithm achieves the desired dependencies on $\epsilon$ and $\delta$, up to polylogarithmic factors, with a slight increase in the accuracy parameter ($\alpha = 4$). Given the significant departure from previous algorithmic approaches required by our work, we hope that our techniques will inspire further progress in this area.

---

### Decision · Program_Chairs · 2024-09-25

**Decision:**

Accept (poster)

**Comment:**

This paper presents a new algorithm for a natural distribution learning problem that has been well-studied in the theory ML literature. It obtains the first linear time algorithm (linear in the number of choice distributions) that achieves a constant factor approximation (in fact, the best known approximation for a proper learner). All of the reviewers agreed that this was an interesting result, even if the paper falls slightly short of fully resolving the problem (it has a sub-optimal dependence on the statistical error, epsilon). There was also concern that the paper is no sufficiently polished, but we believe the author discussion period will help the authors improve their draft of the paper, so are recommending acceptance.